# Finding patterns in lung cancer protein sequences for drug repurposing

**Belén Otero-Carrasco**[1,2]*, **Paloma Tejera Nevado**[1,2], **Rafael Artiñano Muñoz**[1,2], **Gema Díaz Ferreiro**[2], **Aurora Pérez Pérez**[2], **Juan Pedro Caraça-Valente Hernández**[2], **Alejandro Rodríguez-González**[1,2]*

**1** Centro de Tecnología Biomédica, Universidad Politécnica de Madrid, Pozuelo de Alarcón, Madrid, Spain, **2** ETS Ingenieros Informáticos, Universidad Politécnica de Madrid, Boadilla del Monte, Madrid, Spain.

* alejandro.rg@upm.es

## Abstract

Proteins are fundamental biomolecules composed of one or more chains of amino acids. They are essential for all living organisms, contributing to various biological functions and regulatory processes. Alterations in protein structures and functions are closely linked to diseases, emphasizing the need for in-depth study. A thorough understanding of these associations is crucial for developing targeted and more effective therapeutic strategies.Computational analyses of biomedical data facilitate the identification of specific patterns in proteins associated with diseases, providing novel insights into their biological roles. This study introduces a computational approach designed to detect relevant sequence patterns within proteins. These patterns, characterized by specific amino acid arrangements, can be critical for protein functionality. The proposed methodology was applied to proteins targeted by drugs used in lung cancer treatment, a disease that remains the leading cause of cancer-related mortality worldwide. Given that non-small cell lung cancer represents 85–90% of all lung cancer cases, it was selected as the primary focus of this study. Significant sequence patterns were identified, establishing connections between drug-target proteins and proteins associated with lung cancer. Based on these findings, a novel computational framework was developed to extend this pattern-based analysis to proteins linked to other diseases. By employing this approach, relationships between lung cancer drug-target proteins and proteins associated with four additional cancer types were uncovered. These associations, characterized by shared amino acid sequence features, suggest potential opportunities for drug repurposing. Furthermore, validation through an extensive literature review confirmed biological links between lung cancer drug-target proteins and proteins related to other malignancies, reinforcing the potential of this methodology for identifying new therapeutic applications.

**Data availability statement:** All relevant data are within the manuscript and its Supporting Information files.

**Funding:** This research was funded by the project "Data-driven drug repositioning applying graph neural networks (3DR-GNN)" (PID2021-122659OB-I00) from the Spanish Ministerio de Ciencia e Innovación and Drug repurposing hypotheses through a data-driven approach (GRENADA)" (PDC2022-133173-I00) from the Spanish Ministerio de Ciencia e Innovación. Belen Otero Carrasco's work is supported by "Formación de Personal Investigador" grant (FPI PRE2019-090912) as part of the project "DISNET (Creation and analysis of disease networks for drug repurposing from heterogeneous data sources applied to rare diseases)" (RTI2018-094576-A-I00) from the Spanish Ministerio de Ciencia, Innovación y Universidades. The funders had no role in study design, data collection and analysis, decision to publish, or preparation of the manuscript.

## 1. Introduction

Proteins, made up of 20 amino acids, are crucial for biological functions like structure and catalysis. Their structure is organized into four levels: the primary structure is the linear sequence of amino acids; the secondary involves shapes like alpha helix and beta sheets form through hydrogen bonds; the tertiary structure is the 3D shape resulting from protein folding; and the quaternary structure occurs when multiple polypeptide chains (subunits) combine into complex structures [1]. Protein sequences are essential for understanding diseases as they regulate cellular functions, gene expression, and immune responses. Identifying important regions within these sequences helps uncover disease mechanisms, detect patterns, mutations, or genetic variants, and develop targeted treatments. By studying protein sequences, research can identify therapeutic targets and design drugs to counter abnormal processes. Detecting conserved patterns in related proteins aids in understanding their structure and function, which is particularly important for cancer research, especially lung cancer, the leading cause of cancer deaths worldwide. In 85% of lung cancer cases, the disease develops because of smoking. In addition, most diagnoses are made at an advanced stage, which makes treatment difficult. Screening high-risk individuals can improve early detection and survival rates. The World Health Organization (WHO) (https://www.who.int) emphasizes primary prevention, such as tobacco control and reducing environmental hazards, to lower lung cancer cases. Lung cancer includes two main types: non-small cell lung cancer (NSCLC), which is more common and slow-growing, and small cell lung cancer (SCLC), which progresses rapidly. Globally, lung cancer is the leading cause of cancer deaths, responsible for 1.8 million fatalities in 2022 [2]. Other risk factors include second-hand smoke, occupational hazards, air pollution, genetic factors, and chronic lung conditions.

Non-small cell lung cancer (NSCLC) makes up 85–90% of lung cancer cases [3], with adenocarcinoma (AC) and squamous cell carcinoma (SCC) being the most common subtypes. While histological and immunohistochemical analyses aid in diagnosis, molecular profiling is essential for treatment planning, revealing key alterations in genes like EGFR, KRAS, and ALK, especially in AC patients. Treatment options for NSCLC include surgery, chemotherapy, immunotherapy, and targeted therapies, which are personalized for each patient outcome [4]. Targeted therapies, such as monoclonal antibodies and tyrosine kinase inhibitors, are especially effective for advanced or metastatic cases, improving patient outcomes. Lung cancer treatments provide a solid foundation for exploring new therapies through drug repositioning (DR) strategies, which have gained importance in recent years [5]. Drug repurposing involves using existing drugs to treat diseases for which they were not originally developed [6]. This approach saves time, reduces costs, and offers greater safety, as these drugs have already been evaluated in patients. It has a success rate of over 30%, compared to just 2% in new drug development [7]. Computational methods further enhance the potential for finding faster, more effective treatments for various unresolved diseases. Among the most promising computational strategies for drug repurposing, network-based approaches have gained significant

attention due to their ability to model complex interactions between drugs, targets, and disease pathways. Recent studies have demonstrated the effectiveness of these methods in identifying potential therapeutic agents by leveraging network topology and multi-omics data integration [8–11]. Their application has provided relevant insights into disease mechanisms and drug action, complementing other computational methodologies.

In this context, computational drug repurposing approaches have gained attention in recent years and utilize databases that enable gene and protein function prediction by comparing amino acid sequences, with tools like BLAST and FASTA [12]. Mutations in the DNA's protein-coding regions are often linked to human genetic disorders, providing insights into disease mechanisms through protein structure analysis. Understanding disease severity requires identifying specific base-pair mutations, which vary based on protein function, the number of affected amino acids, and mutation type. For example, base pair substitutions can cause silent mutations, while insertions or deletions may result in frameshift mutations, potentially resulting in nonfunctional proteins. Additionally, mutations involving multiples of three base pairs can affect protein functionality differently [13].

The proposed shift in genetic disease classification focuses on molecular pathways rather than traditional categories like monogenic, oligogenic, or polygenic/multifactorial. This pathway-based system organizes disease according to the affected molecular pathways that produce specific phenotypes, enhancing our understanding of disease presentation and progression [14]. Additionally, the classification highlights correlations between disease-related proteins and the sequences of charged residues, aiding in function determination and evolutionary tracking. Effective communication between distant residues is vital for protein functionality, with bioinformatics tools helping identify correlated residues. There is also growing interest in how certain sequence characteristics may lead to protein aggregation in disease contexts [15]. Finally, pathological classification categorizes diseases based on their characteristics, such as cancer (uncontrolled cell growth) and inflammatory diseases (autoimmunity).

Computational methods are crucial in analyzing protein function, offering a precise and efficient approach to understanding complex protein mechanisms [16]. These methods process large amounts of biological data from genomics, proteomics, and structural biology. Using advanced algorithms and modeling, researchers can identify functional motifs, protein domains, and interactions [17]. Moreover, computational tools are key in predicting protein structures and their dynamic behaviors, providing insights into protein folding, assembly, and conformational changes. This understanding is essential for studying the effects of genetic variations, post-translational modifications, and environmental influences on protein function and regulation [18].

Detecting significant patterns within protein amino acid sequences is of great biological interest, leading to various studies utilizing computational techniques to achieve this aim. In [19] the authors employed machine learning-based approaches to predict structural and functional motifs in protein sequences, using protein language models to encode the sequences along with evolutionary information and physicochemical parameters. Another research developed an algorithm based on Teaching Learning-Based Optimization (TLBO) combined with a specialized local search function to identify common patterns in protein sequence datasets [20]. Additionally, a different study outlines a method for locating functionally or structurally critical sites within proteins using sequence data from aligned protein families. This method relies on sequence conservation features within subfamilies and evaluates associations between sites, correcting for phylogenetic bias in the sequence collection [21].

Amino acid sequences play a crucial role in computational techniques for drug repositioning, driving numerous studies in this field. Identifying protein features that improve drug repurposing accuracy is essential for advancing drug design. The CANDO platform has shown success by incorporating a wide variety of indications, diseases, compounds, and proteins to predict interactions based on proteomic signatures [22]. The NOD web server also aids drug repurposing by using a sequence-guided method to predict new drug applications [23]. Additionally, sequence-based drug design introduces a novel approach using protein sequence data through end-to-end differentiable learning [24]. Structural analysis of drug-target-indication relationships further reveals systematic opportunities for drug repurposing [25]. These studies

underscore the value of various computational methods in identifying unique protein patterns to enhance drug design strategies.

In this study, we aim to propose a new drug repurposing strategy based on identifying significant patterns within protein amino acid sequences. These patterns are defined as subsequence's that commonly appear in the target proteins of a drug. Our work aims to find key patterns in the target proteins of drugs used to treat a specific disease, and those same patterns appear in proteins linked to other diseases, which may indicate potential drug repositioning. To explore this, we first select treatments for a specific disease and search for characteristic patterns, followed by a biological interpretation of their relevance to the disease and drug action.

## 2. Materials and methods

### 2.1 Selection of diseases

The primary aim of this study is to propose that the key patterns identified in drug target proteins for NSCLC treatment can be used as a foundation for discovering new opportunities for drug repurposing. To achieve this, a set of diseases should be selected to evaluate these patterns, and their relevance, if detected, can be further analyzed.

The relevant patterns found within the drug's target proteins for NSCLC were first searched for within the sequences of proteins specific to NSCLC. This analysis aimed to determine whether the patterns relevant to lung cancer treatments also applied to the disease itself.

After completing this search, the same analysis was proposed for other types of cancer. It is widely recognized that various cancers exhibit commonalities in their gene and protein expression [26]. Based on this premise, four types of cancer (breast, colon, pancreas, and head and neck) were selected, and all protein sequences associated with these diseases were obtained. The selection of these cancers was based on data from the National Cancer Institute (NIH) (https://seer.cancer.gov/statistics-network/), which indicates that lung and bronchus, colorectal, pancreas, and breast cancers account for nearly 50% of all cancer-related deaths in the USA. Furthermore, breast cancer was included because it is the most diagnosed cancer worldwide [27], with an estimated 310,720 women and 2,790 men expected to be diagnosed in 2024 [28]. It is the second leading cause of cancer-related deaths in women, following lung cancer. Despite a decreasing mortality rate and a 5-year survival rate of 91.2%, its prevalence increases by approximately 0.5% annually [29]. Colorectal cancer was selected because, according to the WHO, it ranks as the third most common cancer globally and is the second leading cause of cancer-related deaths [30]. It mainly affects older individuals and is influenced by lifestyle factors such as diet, exercise, obesity, smoking, and alcohol consumption, with diagnosis often occurring late, limiting treatment options [31]. Pancreatic cancer was chosen due to its high mortality rate and aggressive nature [32], with a 5-year survival rate of around 12% [33]. Early detection is difficult, as symptoms are often absent or non-specific, and the pancreas is hidden behind the organs. It primarily affects older people, with most cases diagnosed after age 75 and rarely occurring in individuals under 40 [34]. Lastly, head and neck cancer was introduced because, though rare, it accounts for about 4% of all cancer cases in the United States, with a higher prevalence among men and older individuals. In 2021, 68,000 cases were diagnosed, mostly involving the mouth, throat, or larynx, while cancers of the paranasal sinuses, nasal cavities, and salivary glands are less common [35].

### 2.2 Datasets collection and preparation

To identify significant patterns within the protein sequences of treatments, we first collected the amino acid sequences corresponding to each drug target protein for lung cancer. The primary source of this information was DISNET [36], a biomedical knowledge platform that consolidates data on various diseases. DISNET compiles information from public databases, incorporating both structured and unstructured data. It is organized into three layers: the phenotypic layer, which focuses on disease-symptom associations; the biological layer, which provides data on genes, pathways, proteins, and

other biological elements and their connections to diseases; and the pharmacological layer, which contains information on medications, including their associations with diseases and drug targets.

In DISNET, the disease non-small cell lung cancer (UMLS ID: C0007131) was identified. Once the disease was chosen, as discussed in previous sections, all relevant information related to it for this study was extracted. Initially, a dataset containing 931 records related to genes, proteins, and associated sequences for non-small cell lung cancer was obtained. Of these 931 records, 923 corresponded to unique proteins. Since some identical proteins were listed under secondary names in the literature, the first step was to filter the 'protein_id' from each record, using a mapping from UniProt. This process removed identical proteins registered under different names, considering the primary and secondary accessions. Additionally, duplicate protein sequences were discarded. After applying this filter, 922 proteins associated with lung cancer were identified. A further filtering step was conducted to check if any of these proteins were target proteins for the lung cancer treatments under consideration. This step eliminated 20 proteins, resulting in a final dataset of 902 unique proteins, referred to as "Disease_lung" throughout the manuscript.

Secondly, all drugs linked to the disease in DISNET were collected as described previously. After filtering the drugs, the related target proteins and their amino acid sequences were determined, along with the corresponding genes. This dataset includes 52 distinct target proteins, referred to as "Treatment_lung" throughout the manuscript.

In contrast, a third dataset, "Full_protein_lung" was created, combining all proteins from the first two datasets. After filtering out overlapping entries, 954 distinct proteins for lung cancer remained (52 from treatment and 902 from lung cancer). Proteins with different names but identical sequences, according to UniProt, were also removed.

A new dataset was then compiled, encompassing the amino acid sequences of proteins involved in four cancers: breast, colon, pancreas, and head and neck. Initially, containing 5,960 proteins, the dataset was reduced to 5,890 after UniProt filtering. After eliminating lung cancer target proteins, the final count was 5,812 proteins, called "Cancers". Some proteins in this dataset are duplicated due to their association with multiple cancer types (see Fig 1).

**2.2.1 Drugs for lung cancer.** Non-small cell lung cancer is a condition characterized by a wide range of treatment options. For this study, we examined all treatments listed as "therapeutic" drugs for this disease within the DISNET knowledge platform. This designation signifies that the drug is used to manage the disease's symptoms. The therapeutic classification was initially sourced from the Comparative Toxicogenomics Database (CTD) and was further refined by including only those treatments for which drug target protein information was available within the platform. Based on these criteria, we identified 43 distinct drugs for lung cancer. These drugs interact with 52 unique target proteins, resulting in 74

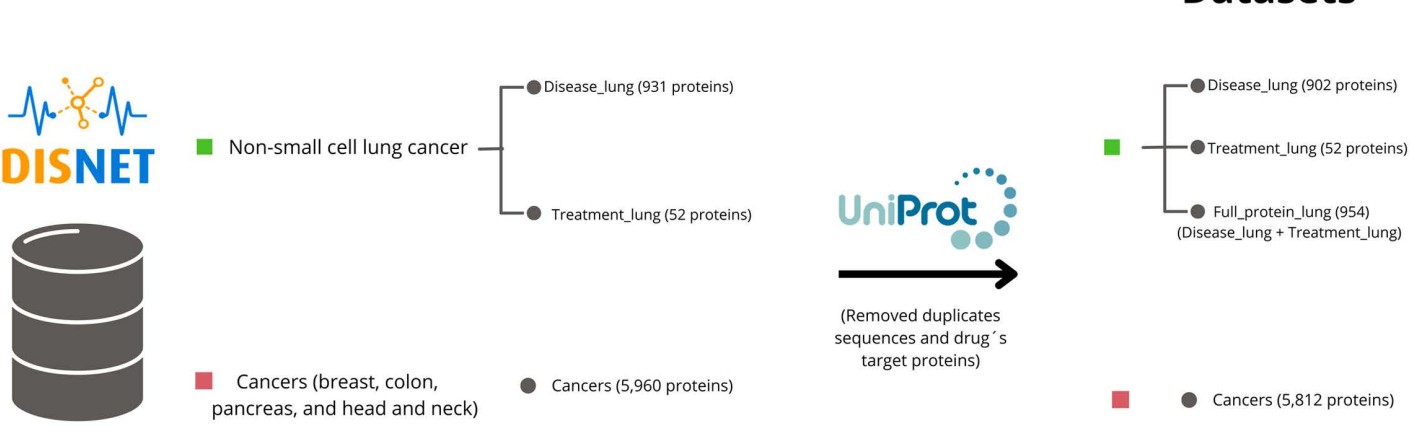

**Fig 1. Summary of the final datasets that will be used throughout the study.**

possible drug-target protein combinations, as some drugs affect multiple target proteins. S1 Table in the supplementary material provides a detailed list of these drugs, their target proteins, corresponding genes, and the drug's effects on the disease.

## 2.3. Patterns

The process of searching for patterns within the amino acid sequences of target proteins associated with non-small cell lung cancer (NSCLC) treatments was carried out. Additionally, the identified patterns were analyzed to determine if they were present in protein sequences from various other diseases. Patterns are defined as substrings of amino acids that appear in at least a minimum number of protein sequences within the dataset. The position of the substring within the protein sequence is not relevant; it can occur anywhere in the sequence to qualify as a pattern. For example, a protein sequence represented as a string of letters, where each letter corresponds to an amino acid, may contain a substring, which is a continuous segment of that string. The discovery of patterns was influenced by both the number of protein sequences analyzed and the length of the proteins examined. In this study, patterns contained within other patterns were excluded to focus on the most informative ones. By adjusting the minimum number of sequences required to contain a pattern, the number of identified patterns can be controlled, allowing for the establishment of specific thresholds.

Initially, identical patterns were searched for within the target proteins of NSCLC treatments. The pattern search was conducted with a minimum occurrence requirement of 5% or 10% of the NSCLC target proteins. These thresholds were selected based on specific criteria outlined in detail in S1 File Methods of the Supporting Information, where we provide a comprehensive explanation of the rationale behind their application in relation to the data set and the objectives of the analysis. This occurrence threshold ensures that a candidate pattern must appear in a certain percentage of proteins to be considered relevant. The process is illustrated in Fig 2. The combination of the occurrence rate and the nature of the

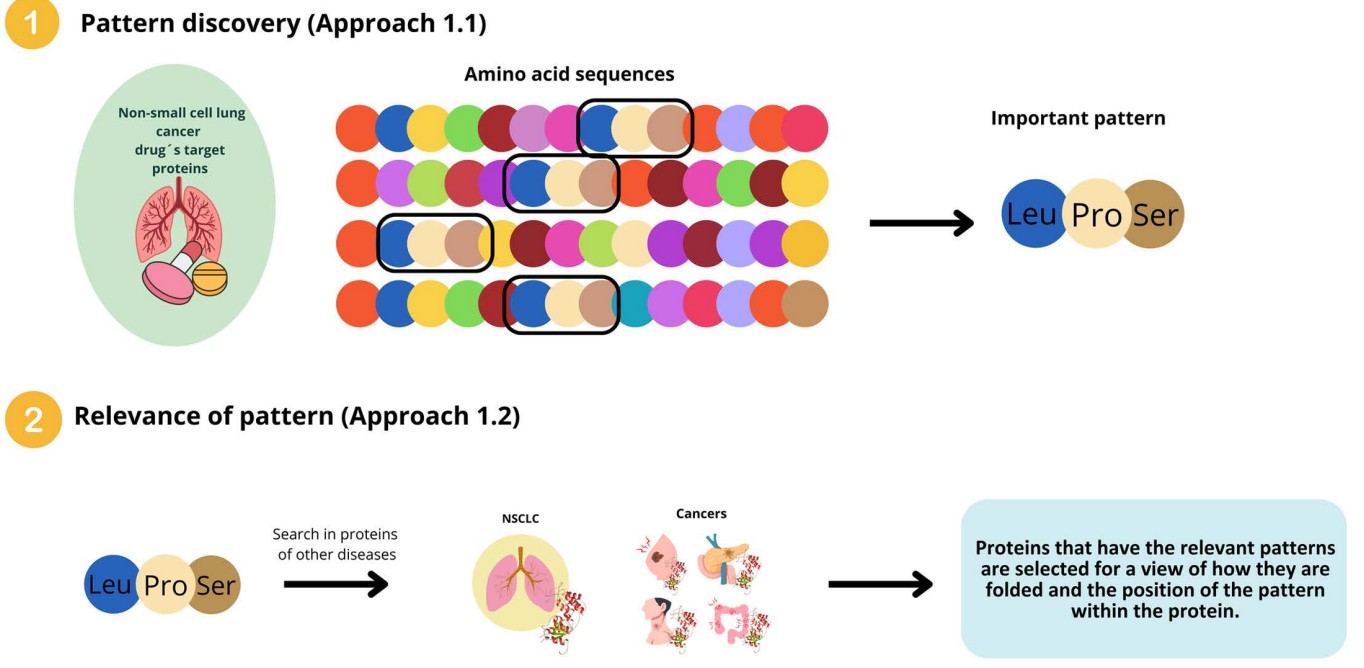

**Fig 2. Overview of the methodology designed to detect significant patterns within NSCLC drug target proteins, aimed at identifying potential opportunities for drug repositioning.**

proteins determines the length and number of patterns identified. The approach focuses on finding the longest possible patterns by progressively extending shorter patterns and checking if they continue to meet the occurrence threshold in a sufficient number of proteins.

The goal of this second step is to identify the longest patterns within the treatment proteins. Afterward, the objective shifts to determining whether these treatment protein patterns are also present in proteins from different diseases. This approach is divided into two parts: the pattern discovery process described in Fig 3, and the search for the previously identified treatment target protein patterns within disease-related proteins, as outlined in Fig 4.

The patterns identified using the proposed approach 1.1, applied to the lung cancer treatment dataset, represent potentially significant structures within the set of treatment proteins. It is hypothesized that these structures could be valuable for drug repositioning, based on the assumption that treatments targeting specific proteins may apply to other proteins with similar patterns, indicating common functional points shared across different proteins.

As previously mentioned, the occurrence value represents the minimum percentage of protein sequences in the dataset that must contain a subsequence of consecutive amino acids for it to be considered a pattern. In this study, the subsequence's that meet these criteria were designated as patterns. During the pattern search, the algorithm generates iterations from patterns of length 1 to the longest subsequence that satisfies the occurrence condition, allowing for the

---

**Approach 1.1** Finding patterns in the drug´s target proteins sequences of NSCLC treatments

1: **procedure PatternIdentification** (TreatmentSequences, %ocurrence)
2:   Patterns ← ∅
3:   ocurrence ← num(TreatmentSequences) * %ocurrence
4:   **for** sequence ← TreatmentSequences **do**
5:     **for** position, amino acid ← sequence **do**
6:       Update(Patterns,amino acid,sequence,position)
7:   **if** number_of_sequences(amino acid) < %ocurrence **then**
8:     removePatternsLenghtI(amino acid,Patterns)
9:   **for** $i \leftarrow 2$ **to** maxPatternLenght **do**
10:     PatternsLenghtI ← getPatternswithsize(Patterns,$i-1$)
11:     **for** pattern,sequences,positions ← PatternsLenghtI **do**
12:       **for** sequence ← sequences,position ← positions **do**
13:         **if** position $+ (i-1)$ < size(sequence) - 1 **then**
14:           Update(Patterns,pattern + new amino acid, sequence,position)
15:     **for** newPattern,sequences ← getPatternswithsize(Patterns,$i$) **do**
16:       **if** size(sequences) < %ocurrence **then**
17:         Delete(Patterns,newPattern)
18:     **for** pattern ← Patterns **do**
19:       **if** IsSubpattern(pattern, Patterns - pattern) **then**
20:         Delete(Patterns,pattern)
21:   **return** Patterns

**Fig 3. The algorithm is designed to identify patterns in the treatment dataset.** This process involves two key functions. First, from lines 1 to 8, the algorithm identifies amino acids that meet the occurrence threshold, starting with patterns of length 1. It then iteratively extends these potential patterns by adding amino acids to previously identified patterns of maximum length. From lines 9 to 21, the algorithm uses the previously found maximum-length patterns to identify those that continue to meet the occurrence criteria and eliminates those that are subpatterns of newly extended patterns (lines 19-21).

elimination of embedded patterns. The occurrence value plays a critical role in determining whether a potential pattern is included in the analysis. Higher occurrence values tend to result in the detection of more general patterns, while lower values lead to more specific patterns.

In this study, the occurrence value significantly influenced the pattern search process. When an occurrence value of 10% was used, shorter patterns were identified compared to when a 5% threshold was applied. As the occurrence value decreases, the likelihood of detecting longer patterns increases. For instance, when the occurrence was set to values higher than 10%, patterns found did not exceed a length of three amino acids, due to the distinct nature of the proteins in the NSCLC treatment dataset. Based on this observation, an occurrence value of 10% or lower yielded better results for identifying relevant patterns. To ensure statistical significance and to minimize the chance of random occurrence, the study required patterns to have a minimum length of four amino acids. Less frequent patterns are more likely to contain meaningful information [37].The justification for this length threshold is thoroughly discussed in S1 File Methods of the Supporting Information, where we detail the criteria used to determine this parameter.

The patterns identified in the NSCLC treatment dataset were then searched for in various disease datasets, including lung cancer and other cancers. One goal of this research was to detect proteins in these diseases that shared the patterns (approach 1.2 (Fig 4)). This matching process was conducted using regular expression (regex) searches to identify proteins containing the specific pattern in any part of their sequence. The search was conducted using patterns with an occurrence rate of 5% and 10%, and all proteins containing each pattern were identified and stored for further statistical and biological analysis. This approach assumes that the sequence structure is maintained regardless of the position of the pattern, though the method can be adjusted if needed to preserve patterns with important structural properties.

After identifying all potential patterns for both occurrences, the goal is to utilize these patterns as a foundation for potential interaction points among disease proteins. To achieve this, we sought patterns that are prevalent in both treatment and disease proteins. The examination of these patterns in NSCLC treatment proteins was carried out across various disease datasets.

We investigated these patterns in two distinct datasets encompassing proteins from various diseases: NSCLC and four different cancer types (breast, colon, pancreas, and head and neck). Significant measures were implemented to exclude any proteins already present in the "Treatment_lung" dataset from these two datasets. This approach enables us to explore different methods and assess the limitations of this technique, which aligns with the optimization efforts aimed at expediting the drug repurposing search. It was discovered that shorter, elongated patterns convey minimal information, indicating that both the quantity and length of patterns correlate directly with the number of proteins within the dataset.

This methodology seeks to provide a fresh perspective on established protein sequence relationships without relying on heavy learning techniques typical of deep learning. However, it retains sufficient relevance to yield insights into related proteins and suggest new avenues for identifying repurposing targets [38]. The proposed method serves as an

---

**Approach 1.2** Searching already found patterns in drug´s target protein sequences of NSCLC treatments in different diseases datasets

```
1:  procedure PatternSearch(DiseasesSequences,Patterns)
        foundPatterns ← ∅
2:      for pattern ← Patterns do
3:          for diseaseSequence ← DiseasesSequences do
4:              if IsSubpattern(pattern, diseaseSequence) then
5:                  Update(foundPatterns, diseaseSequence)
6:      return foundPatterns
```

**Fig 4. The procedure involves verifying whether patterns from NSCLC treatment sequences are present in sequences from various disease datasets, which include lung cancer, and four other cancers (breast, colon, pancreas, and head and neck).**

initial approach for more sophisticated interaction modeling techniques, such as those employing machine learning or laboratory-based experiments [39]. This represents a valuable alternative, as it is lighter [40] and less susceptible to overfitting [38]. Moreover, since the information in the proteins is not altered, it allows for the inclusion of extensive protein structure information in the statistical analysis. This makes it a suitable option for searching within the data structure without modifying the foundational structure. Additionally, this algorithm can be readily extrapolated to conduct more intricate pattern searches [41], enabling it to evolve and become increasingly complex and practical.

The code developed throughout this study can be accessed via the following GitLab link: https://medal.ctb.upm.es/internal/gitlab/b.otero/lung_cancer_finding_patterns

## 2.4 A 3D protein structure analysis pattern

Once relevant patterns in NSCLC treatment proteins were identified among proteins from other cancers, the next step involved verifying whether the disease protein and the target protein sharing the pattern exhibited similar three-dimensional structures. Additionally, this spatial visualization helped pinpoint the specific regions of the proteins where the shared pattern was located and whether this pattern contributed to the same secondary protein structures, such as alpha helices and beta sheets.

Protein sequences were retrieved from UniProt, and Protein Data Bank (PDB) files were obtained from the AlphaFold Protein Structure database [42,43]. The PDB format is the standard for storing files that contain atomic coordinates. Structure visualization was conducted using ChimeraX (v 1.6.1) [44]. The structural comparison was performed using the 'Tools> Sequences Analysis> MatchMaker' function with default parameters, where chain pairing was determined based on the best-aligning pairs of chains between the reference and matching structures. The amino acid sequences representing protein patterns were identified and highlighted to visualize their locations and structural features. The MatchMaker tool facilitates the alignment and matching of residues in three dimensions. It employs secondary structure information to enhance alignment accuracy, particularly for distantly related proteins. The default fitting process excludes dissimilar regions to allow for the close superimposition of similar parts (Fig 5).

In the evaluation of protein structures, various methods are routinely used to compare different protein structures and assess their similarities and differences. RMSD (Root Mean Square Deviation) is the most frequently employed quantitative measure for evaluating the similarity between two superimposed sets of atomic coordinates, typically expressed in angstroms (Å). Global RMSD, calculated based on positional distance, is commonly used to evaluate the overall similarity of protein structures [45]. Specifically, ChimeraX was used to calculate RMSD, which measures the average distance between corresponding atoms of aligned or superimposed molecules. In this study, the reference structure was compared to other proteins using this approach.

## 2.5. Similarity between proteins

Once the relevant patterns within the amino acid sequences of proteins have been detected, knowing the similarity between pairs of proteins that share a pattern can be of interest filter out potential cases of drug repurposing. Calculating protein similarity helps determine how closely related two proteins are by comparing the amino acids in their sequences. In its primary structure, a protein is formed from a linear chain of amino acids, where each amino acid is represented by a single character in the sequence. By assessing the similarity between two proteins, we can better understand their biological relationship. This information is crucial for proposing potential drug repositioning cases, as it enables us to assess whether such cases are more likely to arise between proteins with higher or lower similarity.

To conduct this analysis, a computational method was developed to measure the similarity between protein sequences. This method compares two amino acid sequences and assigns a score based on their lexicographic similarity. The approach also accounts for individual scores for each pair of characters, representing amino acids. The process works

as follows: first, the amino acids are compared, and if they match, they are considered aligned. If they do not match, two scenarios are possible. In the first case, a subsequent series of amino acids may match between the two sequences, resulting in a higher score than treating the initial mismatch as a gap. In the second case, the mismatch is accepted if no better alignment can be identified to improve the overall similarity score (see Fig 6).

The inclusion of gaps and mismatches allows for the comparison of protein sequences of different lengths. Since there is no fixed alignment between the two sequences during comparison, matrix-based scoring systems provide the most effective way to represent this process. In this study, the Needleman-Wunsch (NW) algorithm was used as the foundation, due to its flexibility in assigning weights to matches, mismatches, and gaps. The proposed computational approach modifies this algorithm by introducing custom weights for each pair in the NW input. This modification allows users to tailor the scores for matches, gaps, and mismatches, enhancing the scoring system's adaptability to the specific biological context. The variation in scores, as well as the inclusion of gaps and mismatches, reflects evolutionary, functional, or structural relationships in the similarity analysis [46].

Matching amino acid pairs was prioritized, as the study identifies common patterns. We modified the Needleman-Wunsch (NW) output by normalizing it, producing a numerical score between 0 and 100 to represent the similarity between two protein sequences. The sequences being compared do not need to have the same length, as gaps are introduced to account for differences. This normalization ensures that comparisons between scores are independent of the alignment algorithm used. By standardizing the scores on a 0–100 scale, we create a unified framework that allows us to assess how different amino acids impact the similarity score between two sequences (see Fig 7).

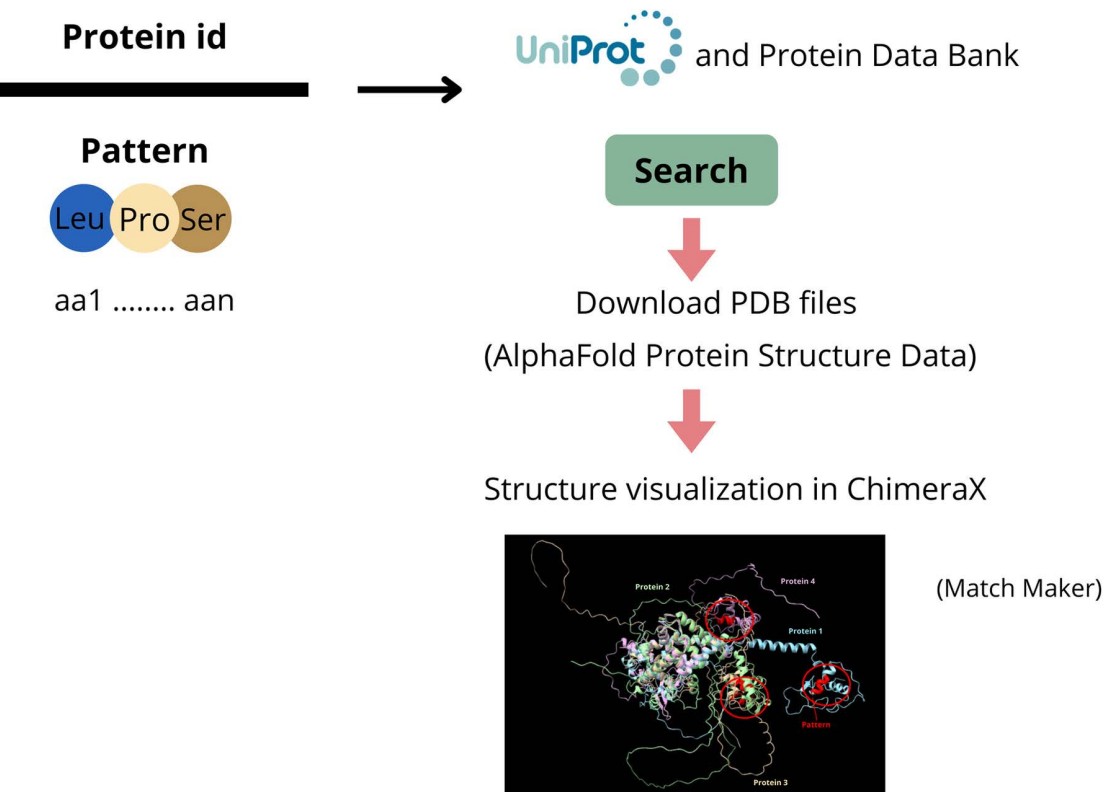

**Fig 5. An overview of the procedure for generating the 3D structure of the proteins and identifying common patterns using ChimeraX.**

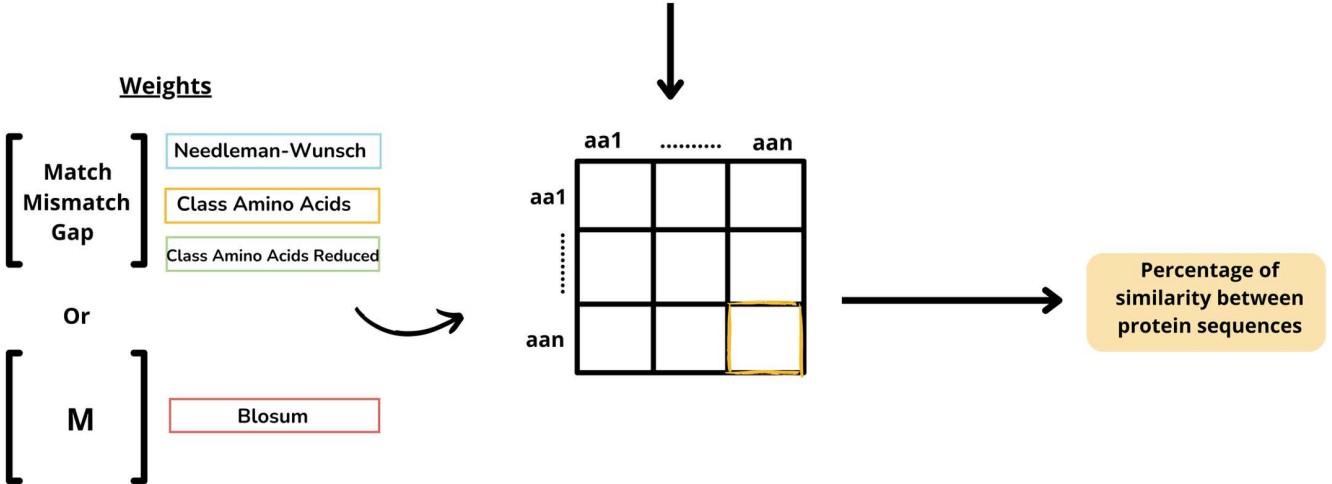

**Fig 6. Methodology used to calculate the similarity between the sequences of the proteins under study, with due consideration of the various metrics.**

The flexibility of the pseudocode allows for the adaptation of custom weights for each amino acid pair. In this case, the scoring for each pair was adjusted according to the specific nature of the protein sequences. Four different strategies were explored, to examine how varying levels of similarity between amino acid pairs affect the results. For each strategy, if no specific score was provided, the default values from the first strategy were used: 3 for a match, 0 for a mismatch, and -1 for a gap. The first strategy, which evaluates strict global similarity based on identical amino acids, provides an agnostic measure of sequence similarity. This initial strategy will be referred to as "Needleman-Wunsch_weights".

The second approach builds on the similarity concept introduced in the first strategy. Its main objective is to account for amino acids that are biologically similar, not just identical. This approach aims to enhance the scoring of sequences that share biological similarities, even when their amino acids do not match exactly. To include this property, a softer ruling was applied, which considers a match for every pair of amino acids that belong to the same group according to Valdar's classification [47]. The amino acids were classified into several categories: hydrophobic (I, L, V, C, A, G, M, F, Y, W, H, K, T), small (V, C, A, G, D, N, S, T, P), aromatic (F, Y, W, H), positive (H, K, R), and negative (E, D). Using this classification, every pair that falls into the same group is considered a match, resulting in significantly more matches than with the simple scoring pairing. This methodology guarantees that information regarding the relationships between amino acids is included. In this context, the establishment of rules allows for the identification of new relationships between pairs of amino acids. A match is defined when both amino acids in a pair belong to the same group, a mismatch occurs when they belong to different groups, and a gap is noted when at least one of the amino acids does not belong to any group. The scores derived from this classification will be referred to as "Class_amino_acids" from this point forward.

Given the high number of amino acid matches identified using the classification established by Valdar [47], a revised classification that takes into account the properties of amino acids was proposed, consisting of the following categories: tiny (A, G, S), aliphatic (I, L, V), aromatic (F, Y, W), positive (H, K, R), and negative (E, D). This new classification is more selective in determining matches. As shown in Fig 7, the previously mentioned gap remains, enabling continued

---

**Approach 2** Similarity scoring approach using custom weights

```
 1:  procedure  Custom_weights_similarity_score  (prot1,  prot2,  pair_scores,
         match, mismatch, gap)
 2:      aa_tab ← (amino acid prot1, amino acid prot2)
 3:      while aa_tab ≠ ∅ do
 4:          aa1, aa2 ← get_pair(aa_tab)
 5:          if aa1 = aa2 or exists(pair_scores(aa1,aa2)) then
 6:              if exists(pair_scores(aa1,aa2)) then
 7:                  Matrix(aa1,aa2) ← pair_scores(aa1,aa2)
 8:              else
 9:                  Matrix(aa1,aa2) ← match
10:          else
11:                  Matrix(aa1,aa2) ← mismatch
12:      for i ← prot1, j ← prot2 do
13:          if i ≠ 0 and j ≠ 0 then
14:              F[i, j] ← 0
15:      for i ← 0 to len(prot1) do
16:          F(i,0) ← gap × i
17:      for j ← 0 to len(prot2) do
18:          F(0, j) ← gap × j
19:      for i ← 1 to len(prot1) do
20:          for j ← 1 to len(prot2) do
21:              Match ← F(i − 1, j − 1) + Matrix(prot1[i], prot2[j])
22:              Delete ← F(i − 1, j) + gap
23:              Insert ← F(i, j − 1) + gap
24:              F(i, j) ← max(Match, Insert, Delete)
25:      return F(len(prot1) - 1, len(prot2) - 1)
```

---

**Fig 7. The pseudocode outlines the proposed method for scoring the similarity between two protein sequences.** The Needleman-Wunsch (NW) alignment algorithm was adapted to allow custom values for each character pair in the alphabet, stored in a matrix-like structure. The process begins by setting up the scores for all amino acid pairs between lines 2 and 11, where values are assigned according to specified rules (pair_scores, line 5). Additionally, values for matches, mismatches, and gaps are also defined. Lines 12 to 18 describe the initialization of the score matrix. Initially, all rows and columns are set to 0, followed by the assignment of gap penalties to column 0 and row 0. The scoring algorithm is then implemented starting at line 19. This method uses a dynamic programming approach to compute the optimal alignment score between the two sequences. During the iteration, the maximum value is selected from among match, delete, and insert operations (lines 21, 22, and 23). The final output is a score that quantifies the similarity between the two input sequences (line 25).

consideration of the size differences among various proteins. The strategy referred to as "Class_amino_acids_reduced" will be used throughout the manuscript.

The metrics described above refer to fixed scores assigned for each possible outcome (match, mismatch, and gap). Consequently, an additional strategy based on the BLOSUM 62 matrix [48] was implemented. This matrix allows for the assignment of a graded score to amino acid pairs assessed along the sequences. The BLOSUM 62 similarity matrix is utilized within the Blast algorithm to detect low similarity between proteins, and it is considered one of the most effective matrices for this purpose. By incorporating BLOSUM 62 alongside the match, mismatch, and gap values used in previous strategies for pairs not covered by BLOSUM, a new set of values is generated, providing more accurate scoring for most potential scenarios included in the dataset. The results obtained from applying these scores are then normalized to ensure that the final values are within the 0–100 range. The calculation of these metrics was conducted using the "Disease_lung" dataset and separately on the "Treatment_lung" proteins dataset. Additionally, the similarity between the

"Treatment_lung" proteins and the proteins associated with various diseases, including, "Disease_lung" and "Cancers", was also computed.

This process compares pairs of characters at corresponding positions in the sequences and applies the appropriate scores based on the selected metric. It also accounts for cases where a character in one sequence is compared to a gap in the other. Besides the two protein sequences, the pseudocode optionally accepts a specific scoring scheme for each amino acid pair and default match, mismatch, and gap values. The approach uses this input to generate a pairwise scoring matrix, which is directly passed to the scoring algorithm, unlike many alignment algorithms that use fixed values.

The similarity calculation involves constructing a scoring matrix (referred to as F in Fig 7), where the algorithm calculates the cumulative similarity score cell by cell. The matrix is initiated in the upper left corner and built progressively until the lower right corner is reached. Each cell is populated with the maximum of three possible values: the score derived from the matrix for the current amino acid pair, the score corresponding to advancing one amino acid in protein 1 while holding in protein 2, or the score corresponding to advancing one amino acid in protein 2 while holding in protein 1. This decision-making process is detailed in line 24 of Fig 7. Upon completing the matrix, the similarity score between both sequences is found in the lower right cell, as indicated in line 25.

### 2.6. Statistical analysis of similarity

A statistical analysis was conducted to determine whether significant differences existed in the similarity values between proteins across different scenarios. Four distinct similarity metrics were used in this analysis: Needleman-Wunsch_weights, Class_amino_acids, Class_amino_acids_reduced, and BLOSUM. Initially, a statistical comparison was performed to evaluate the similarity measures obtained from these metrics when proteins from the "Treatment_lung" dataset were compared with those from the other datasets included in the study.

Before conducting the comparison, the normality of the data – i.e., whether the similarity results from the different metrics followed a Gaussian distribution – was assessed using the Lilliefors normality test [49]. The results revealed that none of the four datasets exhibited a normal distribution. Consequently, the Mann-Whitney U test [50] was employed to compare these similarity measures and determine if significant differences in similarity existed between the proteins from these datasets.

Next, the analysis sought to determine if there was a greater similarity between proteins in the "Treatment_Lung" dataset and those in the "Disease_Lung" dataset compared to the similarity between the "Treatment_Lung" dataset, which contains NSCLC treatment target proteins, and the datasets from other cancers. This investigation aimed to assess whether the treatment target proteins were more closely associated with the disease they were intended to treat or whether they exhibited stronger similarity with proteins from other diseases. This second scenario followed the same methodology as the first: normality was first tested using the Lilliefors test, and based on the results, a Mann-Whitney U test was subsequently performed.

### 2.7 Drug repurposing approach

The identification of characteristic patterns in the target proteins of NSCLC treatments paves the way for a novel drug repurposing strategy. This study proposes that patterns occurring with frequencies of 5% or 10% in the amino acid sequences of NSCLC drug target proteins could be crucial in facilitating the use of these drugs for treating other diseases.

The search for significant patterns in other cancer types has revealed shared structures between lung cancer treatments and each of these conditions. In all instances, triples (Target protein – Drug – New protein target) have been identified where both the original and new proteins share at least one pattern, with similarity levels exceeding 95%. This selection process aims to pinpoint highly similar proteins and explore the role of similarity in identifying potential drug repurposing opportunities. Moreover, it has been noted that drug repurposing cases can arise when the similarity between

proteins is less than 5%. This hypothesis stems from the observation that numerous successful repurposing cases throughout history involve distinctly different elements, such as entirely different diseases.

Furthermore, to assess whether sharing a pattern significantly contributes to successful drug repurposing cases without necessitating similarity between the two proteins, we will also thoroughly investigate cases that could lead to genuinely advantageous drug repurposing outcomes.

This new methodology aims to establish a fresh pathway for drug repurposing, emphasizing the importance of shared patterns between proteins from disparate diseases. Integrating these two approaches will generate a comprehensive list of potential repurposing cases for each disease dataset under investigation.

## 3. Results

### 3.1. Patterns

The pattern search methodology described in the Material and Methods section was utilized to identify a set of relevant patterns for this study. The minimum occurrence threshold was established at 5% and 10% of the datasets. This section details the relevant patterns found within the drug target proteins of NSCLC treatments.

In the case of the 5% occurrence threshold, a total of 9,863 pattern-protein combinations were identified. Within these combinations, 4,170 distinct patterns were found, with lengths ranging from 94 to 3 amino acids. The most prevalent pattern, which appears 9 times, is the 3-amino acid sequence "AEA". Regarding the proteins, there are 52 different proteins represented in these combinations. The number of associated patterns varies significantly among these proteins, with a maximum of 955 patterns linked to a single protein and a minimum of 30. On average, each protein is associated with approximately 189 patterns. Additionally, an important filtering criterion is the identification of patterns consisting of four or more amino acids. This filter is applied to select patterns for further analysis in lung cancer proteins and other datasets. At the 5% occurrence threshold, a total of 2,368 unique patterns consisting of four or more amino acids were identified.

In contrast, at the 10% occurrence threshold, a total of 14,330 pattern-protein combinations were identified. Within these combinations, 2,034 distinct patterns were found. Although this case yielded a higher number of pattern-protein combinations, the number of unique patterns is significantly lower than in the previous case. These patterns vary in length from 8 to 2 amino acids, reflecting a notable decrease compared to the 5% occurrence cases. Two patterns, "HA" and "FM" were identified as the most abundant, each appearing 21 times. Regarding the proteins, 52 different proteins were included in these combinations. The number of associated patterns varies, with a maximum of 987 patterns linked to a single protein and a minimum of 33. On average, each protein is associated with approximately 275 patterns. When filtering for patterns consisting of four or more amino acids, a total of 47 unique patterns of this length were identified.

The subsequent subsections describe the identified patterns and their characteristics concerning occurrences in non-small cell lung cancer, as well as other cancers (breast, colorectal, head and neck, and pancreatic cancers).

**3.1.1. Patterns in Non-Small Lung Cancer (NSCLC).** After identifying relevant patterns with occurrences of 5% and 10% in lung cancer treatments, the next step was to assess how many of these patterns of interest appeared in the proteins linked to NSCLC.

On the identified patterns, 2,368 exhibited a 5% occurrence of lung cancer, with 2,071 of these patterns being detected in the proteins associated with the diseases. When focusing on patterns with a 10% occurrence rate, 46 out of the 47 identified patterns were found within the disease's proteins. Fig 8 illustrates the progression in the number of patterns detected at both 5% and 10% occurrence levels as the corresponding filters were applied. These findings are specific to patterns observed in NSCLC.

**3.1.2. Patterns in other types of cancer.** Additionally, this study expanded the search for significant patterns in the target proteins of lung cancer treatments to other protein datasets from different cancer types. As outlined in the methodology, four cancers were selected for this investigation: breast, colorectal, pancreatic, and head and neck

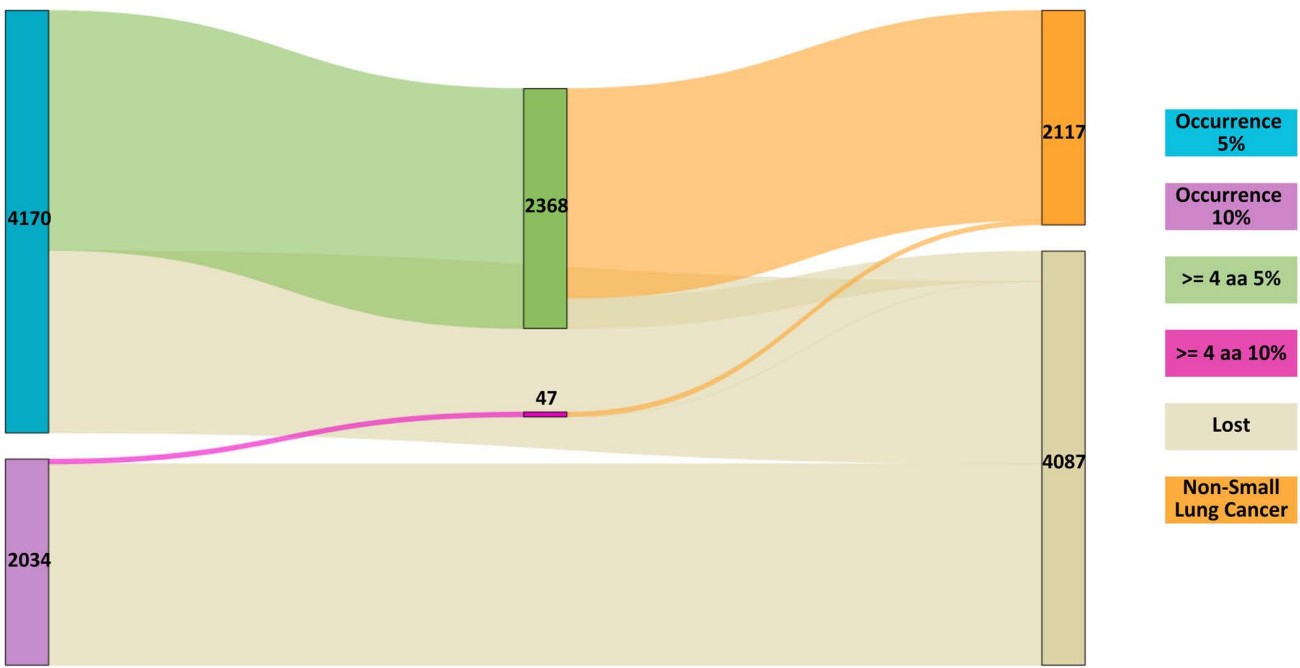

**Fig 8. Sankey plot showing the patterns identified in the amino acid sequences of target proteins for NSCLC drugs, with both 10% and 5% occurrence rates displayed.**

cancers. For each of these diseases, the presence of the characteristic lung cancer treatment patterns with 5% and 10% occurrence rates was verified within their associated proteins.

Among the 2,368 patterns identified with a 5% occurrence in lung cancer treatments containing at least four amino acids, 2,200 were detected in the proteins of various cancer types. A detailed analysis revealed that 2,193 of these patterns appeared in breast cancer proteins, 2,138 in colorectal cancer proteins, 2,023 in pancreatic cancer proteins, and 1,589 in head and neck cancer proteins.

In contrast, of the 47 patterns identified with a 10% occurrence and at least four amino acids, all were present in at least one type of cancer. Specifically, every pattern was detected within the proteins of breast and colorectal cancers. For pancreatic cancer, 46 out of the 47 patterns were found, while in head and neck cancer, 38 patterns were identified within the disease's proteins. Fig 9 shows the distribution of these values across the different stages of the analysis.

### 3.2 Patterns in the 3D structure of proteins

Protein patterns are identified within amino acid sequences based on their similarity, represented as strings of amino acids. Patterns with at least four amino acids were chosen to evaluate whether the developed methodology could effectively identify proteins that share significant structural elements.

Protein structure is defined by four levels. The primary level refers to the linear sequence of amino acids, while the secondary level includes local patterns like alpha helices and beta strands. The tertiary level describes the overall three-dimensional arrangement of amino acids, and the quaternary level focuses on the organization and interactions of multiple protein subunits. Visualization methods, such as space-filling models, depict the overall shape and surface of proteins, whereas ribbon models highlight the elements of secondary structure. To quantify the similarity between protein structures, RMSD was calculated (see section 4.3.3).

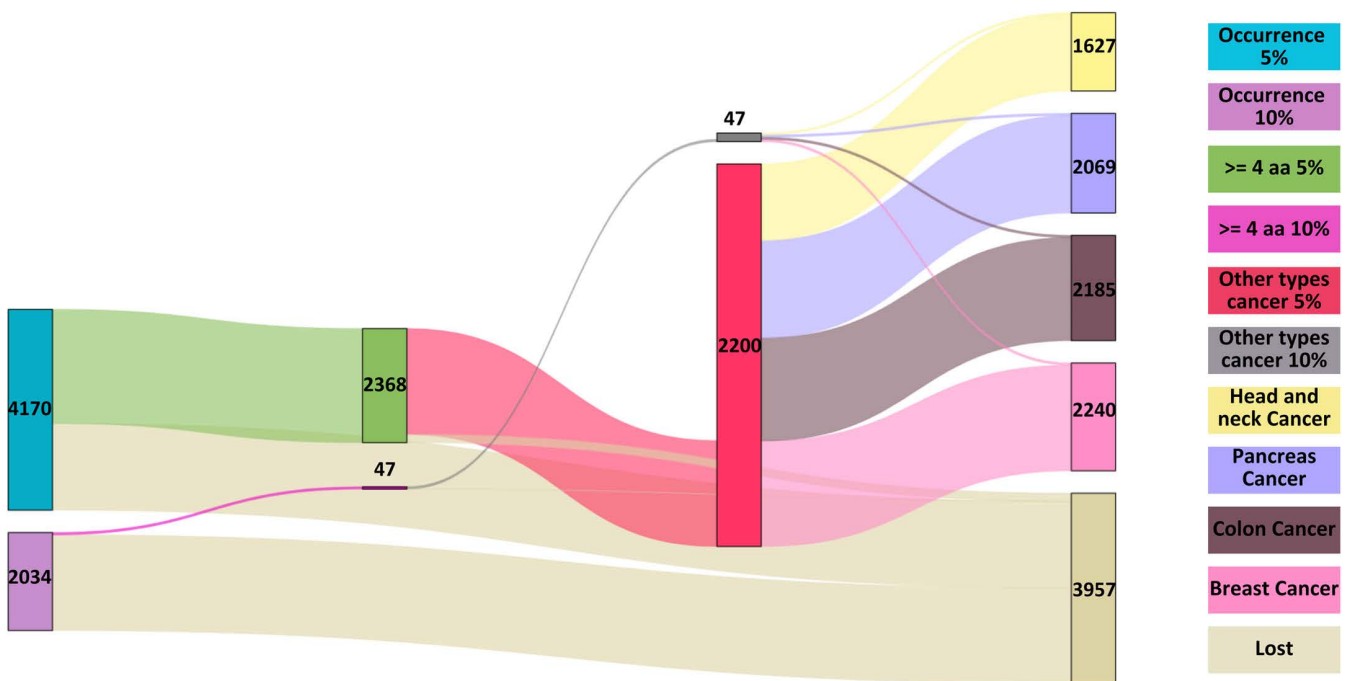

**Fig 9. Analysis of significant NSCLC treatment patterns with 10% and 5% occurrence in the other selected cancer types.**

**3.2.1. Lung cancer.** Proteins displaying patterns that occurred in 10% of cases within the lung cancer dataset were selected for in-depth analysis to identify shared structural features. The pattern 'CEGCKGFF' was detected in the proteins RARB and VDR. A comparative analysis was performed with the proteins NR1I2 and PPARG, which were identified in the treatment dataset (see Fig 10).

Details regarding the protein treatment NR1I2 and the drug targets for RARB and VDR are provided in Table 1.

Paclitaxel (CHEMBL 428647) targets NR1I2 and has similarities to RARB and VDR, although these proteins are not the primary drug targets. There exists a connection involving nuclear receptors (NRs), which are closely associated with cancer due to their essential role in cellular processes related to cancer progression and development. RARB and VDR are specific types of these receptors that have been implicated in cancer.

**3.2.2. Other types of cancer.** Proteins showing patterns with frequencies of both 10% and 5% in the dataset of other cancer types were chosen for further analysis to explore their common structural features (see Fig 11).

The pattern 'CEGCKGFF' was detected with a frequency of 10%. It appeared in different cancer datasets, specifically in THB (P10828), VDR (P11473), NR1D1 (P20393), and RORA (P35398). Additionally, this pattern was found in the protein treatment dataset, which includes NR1I2 (O75469), RARA (P10276), and PPARG (P37231) (see Fig 11 a)). Peroxisome proliferator-activated receptors (PPARs) and thyroid hormone receptors (TRs) belong to subfamily 1 (NR1) of the nuclear receptor superfamily and function as ligand-dependent transcription factors. These receptors interact with their cognate hormone response elements in gene promoters to regulate the expression of target genes, thereby influencing various cellular functions. The NR1 group also comprises retinoic acid receptors (RARs), Reverb, RAR-related orphan receptors (RORs), oxysterol receptors (LXRs), vitamin D3 receptors (VDRs), and the nuclear xenobiotic receptor, also known as the constitutive androstane receptor (CAR). Both PPARs and TRs possess a conserved DNA-binding domain (DBD) and can enhance their activity through heterodimerization with a common partner, the retinoid X receptor (RXR), facilitating the regulation of target gene transcription. PPARs and TRs play significant roles in developmental and metabolic processes

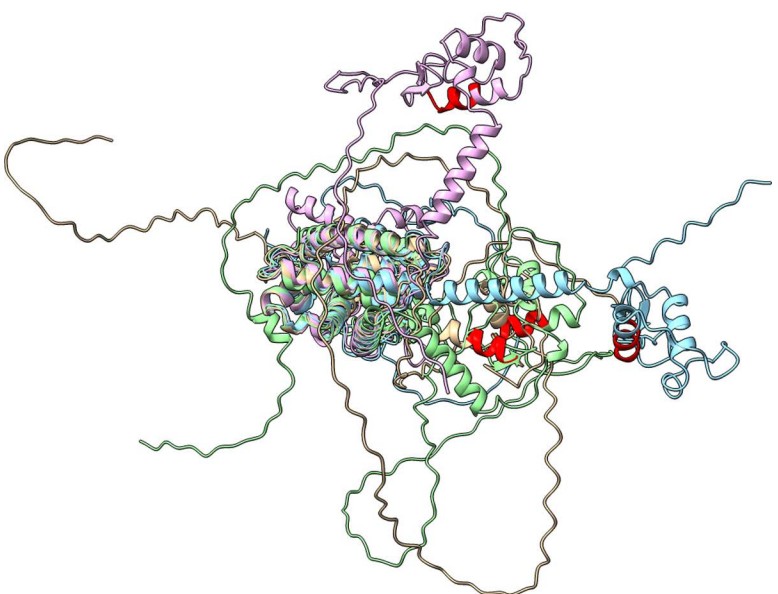

**Fig 10. Superposition of protein structures was performed based on the detected pattern (CEGCKGFF', in red) with a 10% occurrence threshold in the lung cancer dataset.** The reference proteins RARB (P10826, in brown) and VDR (P11473, in blue) were identified in the cancer dataset. For comparison, the proteins from the treatment dataset, NR1I2 (O75469, in pink) and PPARG (P37231, in green), are also highlighted.

**Table 1. Summary of the protein treatments and drug targets from the disease lung datasets. This includes the following details: Protein ID treatment (the unique identifier for each protein), Protein name (the name or description of the protein), Drug treatment (the name of the drug administrated), Drug action (a description of the drug's effect on the protein), Protein ID new (the name of the newly identified protein), and Protein name and Cancer types (specifically, lung cancer).**

| Protein ID treatment | Protein name | Drug treatment | Drug action | Protein ID new | Protein name | Cancer types |
|---|---|---|---|---|---|---|
| O75469 | NR1I2 | Paclitaxel | Inducer | P10826 | RARB | Lung |
| O75469 | NR1I2 | Paclitaxel | Inducer | P11473 | VDR | Lung |

and in diseases like obesity, diabetes, and cancer. All the proteins exhibiting this pattern are receptors that are interconnected at various levels. Isotretinoin (CHEMBL547) specifically targets RARA and shows similarities with the proteins THB, NR1D1, VDR, and RORA, although none of these proteins are drug targets. Retinoic acid (RA) demonstrated anti-tumor activity by promoting cellular differentiation. NR1D1 functions as a nuclear receptor, akin to RARA. Additionally, both RARA and VDR are classified as nuclear receptors that have been linked to cancer.

The pattern 'QKCL' was detected with a 10% occurrence rate. Initially, using the lung cancer dataset, it was identified in VEGFC and proteins from the treatment datasets: RXRA, PPARG, and SMO. Comparing the protein folding predictions among these proteins revealed overlapping structures. Subsequently, the same methodology was applied to a larger dataset covering additional cancer types. In this broader analysis, it was observed that NR4A1, NR4A3, and NR1I3, along with other proteins, exhibit this pattern, with links to RXRA and PPARG in the treatment group. NR4A1, NR4A3, NR1I3, RXRA, and PPARG are nuclear receptors. In contrast, VEGFC belongs to the platelet-derived growth factor/vascular endothelial growth factor (PDGF/VEGF) family, while SMO (Smoothened) is an essential transmembrane protein. Initially, the structure of the VEGFC protein (P49767) was visualized and compared with proteins from the treatment dataset that also contained the pattern. Subsequently, the most relevant proteins from the treatment dataset were aligned to overlap their structures using the Matchmaker tool in ChimeraX (see Fig 11 b) and c)).

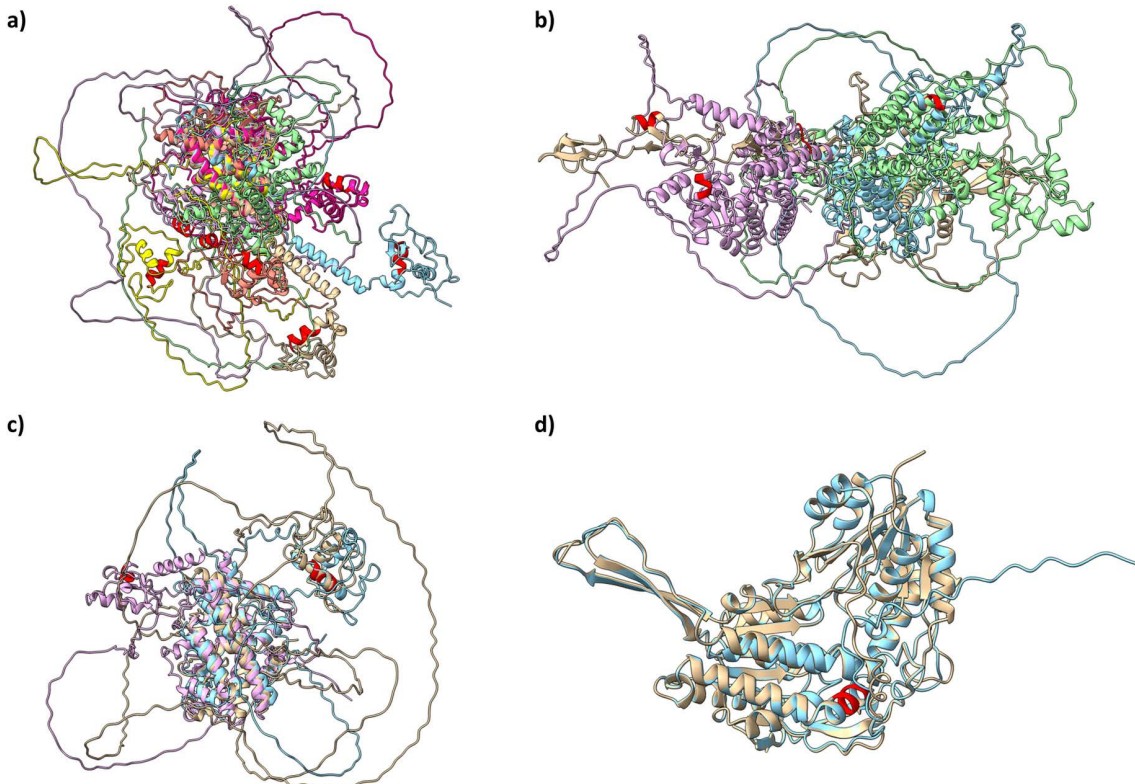

**Fig 11. Superposition of protein structures. a) Protein structures were aligned based on the detected pattern (CEGCKGFF', in red) using a threshold of 10% occurrence in the lung cancer treatment dataset.** In a different cancer dataset, a similar recurring pattern was found in the proteins THB (P10828, in brown), VDR (P11473, in blue), NR1D1 (P20393, in purple), and RORA (P35398, in green). This same pattern was also noted in the protein treatment dataset, which included NR1I2 (O75469, in orange), RARA (P10276, in yellow), and PPARG (P37231, in pink). b) Superposition of protein structures based on the detected 'QKCL' pattern (shown in red), with a 10% occurrence threshold in the lung cancer treatment dataset. The reference protein, VEGFC (P49767, in brown), associated with breast cancer, is compared to proteins from the treatment dataset: RXRA (P19793, in blue), PPARG (P37231, in pink), and SMO (Q99835, in green). c) The reference protein, NR4A3 (Q92570, brown), associated with breast cancer, is compared to proteins from the treatment dataset: RXRA (P19793, in blue) and PPARG (P37231, in pink). d) Protein structures were aligned based on the detected pattern (AALET', in red) with a 5% occurrence threshold from the lung cancer treatment dataset. The reference protein, AL1A3 (P47895, in brown), is linked to breast cancer. For comparison, proteins from the treatment dataset including ALDH2 (P05091, in blue), are highlighted.

The pattern 'AALET' was found with a 5% occurrence rate. It was first identified in the lung cancer dataset within an unnamed protein (NA), which was later confirmed as AL1A3 upon investigation in UniProt (P47895). This pattern was also detected in the ALDH2 (P05091) and MTOR (P42345, not shown) in the treatment dataset (see Fig 11 d).

Details regarding protein treatment and drug targets involved in each different example can be found in Table 2.

**3.2.3 Calculation of RMSD.** The variability in RMSD values allows for comparisons across different protein sequences (Table 3). This quantitative assessment provides a clearer understanding of the structural alignment and deviations between the analysed proteins and their respective reference structures (Figs 10 and 11).

For the pattern 'CEGCKGFF', the RMSD values and the number of residue pairs compared between different structures show the following results. When comparing RARB with VDR, the RMSD is 29.574 Å with 351 pairs. Other comparisons include NR1I2 with an RMSD of 37.546 Å and 367 pairs, and PPARG with an RMSD of 29.616 Å and 412 pairs. Additionally, when looking at the structure matched with THB against VDR, the RMSD is 25.812 Å with 368 pairs. Further comparisons include NR1D1 at 43.557 Å and 439 pairs, RORA at 37.702 Å with 409 pairs, NR1I2 at 24.307 Å with 378 pairs, RARA at 40.824 Å with 408 pairs, and PPARG at 48.713 Å with 440 pairs.

The pattern 'QKCL' shows an evaluation with VEGFC and RXRA, resulting in an RMSD of 31.021 Å and 87 pairs, with additional comparisons such as PPARG showing a higher RMSD of 56.803 Å with 173 pairs, and SMO at lower RMSD of 11.592 Å with 45 pairs. For the same pattern detected, NR4A3 and RXRA show an RMSD of 43.743 Å with 449 pairs, while PPARG exhibits an RMSD of 45.832 Å with 452 pairs.

Lastly, the pattern 'AALET' is compared with AL1A3 and ALDH2, resulting in a notably low RMSD of 4.287 Å with 512 pairs.

### 3.3  Similarities between proteins

This section presents the results of similarity calculations using the four metrics considered in this study: "Needleman-Wunsch_weights", "Class_amino_acids", "Class_amino_acids_reduced" and "Blosum". These calculations were applied to each of the datasets: "Disease_lung", "Treatment_lung", "Full_protein_lung" and "Cancers".

**Table 2. Overview of the protein treatment and drug targets derived from the cancer datasets.**

| Pattern | Protein ID treat | Protein name | Drug Treatment | Drug action | Protein ID new | Protein name | Cancer types |
|---|---|---|---|---|---|---|---|
| 'CEGCKGFF' | P10276 | RARA | Isotretinoin | Unknown | P10828 | THB | Breast |
| | P10276 | RARA | Isotretinoin | Unknown | P20393 | NR1D1 | Breast |
| | P10276 | RARA | Isotretinoin | Unknown | P11473 | VDR | Breast, Colon, Pancreas |
| | P10276 | RARA | Isotretinoin | Unknown | P35398 | RORA | Colon |
| 'QKCL' | P19793 | RXRA | Bexarotene | Agonist | Q92570 | NR4A3 | Breast |
| | P28702 | RXRB | Bexarotene | Agonist | Q92570 | NR4A3 | Breast |
| | P48443 | RXRG | Bexarotene | Agonist | Q92570 | NR4A3 | Breast |
| | P37231 | PPARG | Troglitazone | Agonist | Q92570 | NR4A3 | Breast |
| | Q99835 | SMO | Vismodegib | Inhibitor | Q92570 | NR4A3 | Breast |
| 'AALET' | P05091 | ALDH2 | Disulfiram | Inhibitor | AL1A3 | P47895 | Breast |
| | P42345 | MTOR | Everolimus | Inhibitor | AL1A3 | P47895 | Breast |

**Table 3. Calculation of the RMSD by comparing the reference structure with the target structures.**

| Pattern | Reference Structure | | Structure to match | | RMSD | Description |
|---|---|---|---|---|---|---|
| Amino acid sequence | Protein 1 | Uniprot ID1 | Protein 2 | Uniprot ID2 | Ansgstroms (Å) | Number of pairs |
| CEGCKGFF | RARB | P10826 | VDR | P11473 | 29.574 | 351 |
| | | | NR1I2 | O75469 | 37.546 | 367 |
| | | | PPARG | P37231 | 29.616 | 412 |
| CEGCKGFF | THB | P10828 | VDR | P11473 | 35.812 | 368 |
| | | | NR1D1 | P20393 | 43.557 | 439 |
| | | | RORA | P35398 | 37.702 | 409 |
| | | | NR1I2 | O75469 | 24.307 | 378 |
| | | | RARA | P10276 | 40.824 | 408 |
| | | | PPARG | P37231 | 48.713 | 440 |
| QKCL | VEGFC | P49767 | RXRA | P19793 | 31.021 | 87 |
| | | | PPARG | P37231 | 56.803 | 173 |
| | | | SMO | Q99835 | 11.592 | 45 |
| QKCL | NR4A3 | Q92570 | RXRA | P19793 | 43.743 | 449 |
| | | | PPARG | P37231 | 45.832 | 452 |
| AALET | AL1A3 | P47895 | ALDH2 | P05091 | 4.287 | 512 |

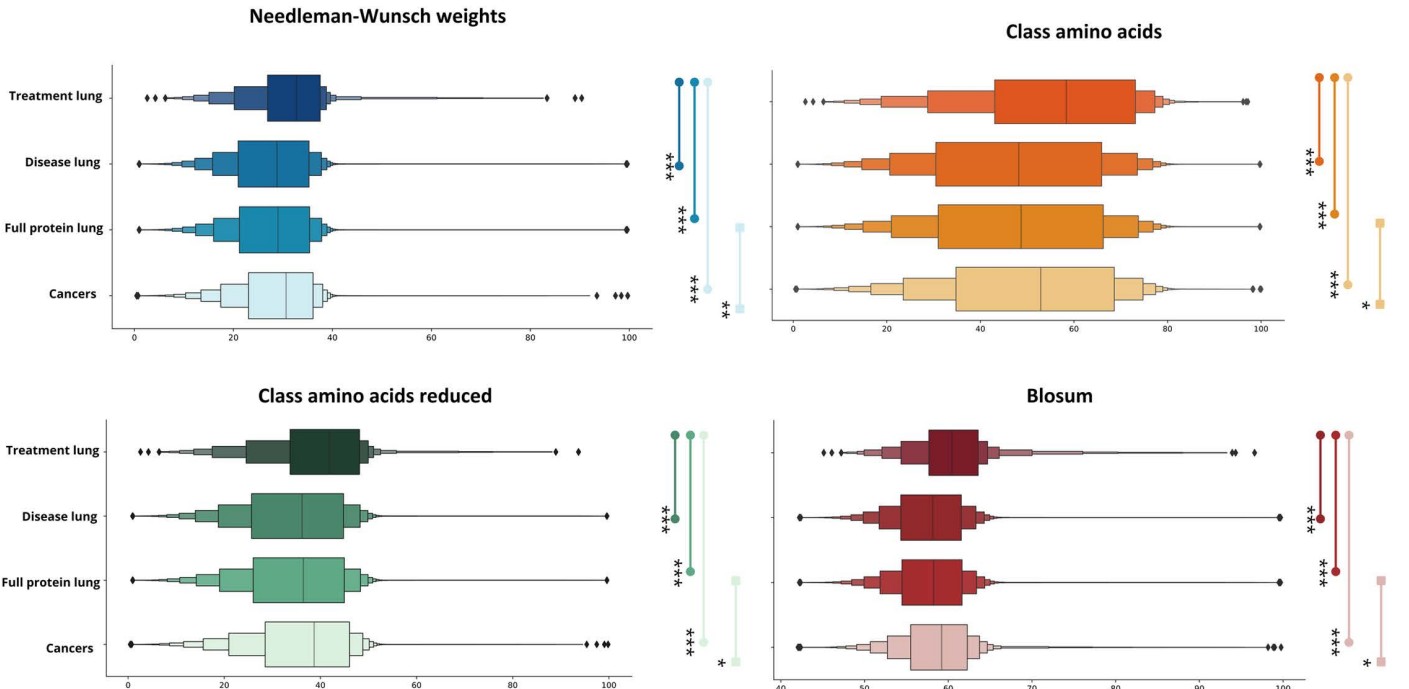

**Fig 12. Boxplot showing the distribution of protein similarity across datasets using the different similarity metrics.** The results of the statistical test for both considered scenarios are also indicated. P-value annotation legend: *: $1.00 \times 10^{-2} < p <= 5.00 \times 10^{-2}$, **: $1.00 \times 10^{-3} < p <= 1.00 \times 10^{-2}$ and ***: $1.00 \times 10^{-4} < p <= 1.00 \times 10^{-3}$.

**Table 4. Mean Protein Similarity and Standard Deviation Across Datasets Using Four Metrics for each dataset.**

| Datasets | Needleman-Wunsch_weights | Class amino_acids | Class_amino acids_reduced | Blosum |
|---|---|---|---|---|
| Disease_lung | 27.55±9.06 | 47.46±20.71 | 34.52±12.01 | 57.74±4.83 |
| Treatment_lung | 31.35±9.01 | 55.80±19.70 | 39.52±11.53 | 60.42±5.48 |
| Full_protein_lung | 27.71±9.01 | 47.83±20.66 | 34.74±11.95 | 57.86±4.85 |
| Cancers | 28.81±8.60 | 50.55±20.28 | 36.30±11.54 | 58.70±4.92 |

We began by analysing the protein similarities within each dataset based on the four-similarity metrics. Table 4 shows the average similarity values for each dataset across all metrics. The similarity is expressed as a percentage, ranging from 0 to 100, where 100 indicates the highest possible similarity between two proteins.

Additionally, the similarity values for each metric across the datasets have been plotted. S1 Fig in supplementary material presents these distributions as histograms, with a separate histogram for each dataset showing the results of the four metrics.

After analyzing the distribution of protein similarity within each dataset, it became essential to assess whether statistically significant differences existed among the datasets. As shown in the similarity distribution table, the "Treatment_lung" dataset appears to have on average, higher protein similarity than the other datasets. To further explore these differences and determine their statistical significance, a Mann-Whitney U test was conducted, comparing the similarity in the "Treatment_lung" dataset with the other datasets. Since four different metrics were used to measure protein similarity, the analysis was performed separately for each metric. Additionally, a comparison was made to assess whether the similarity between NSCLC treatment proteins and NSCLC was greater than that between NSCLC treatment proteins and proteins from other cancer types.

| | | | TRIPLES |
|---|---|---|---|
| **OCURRENCCE 5%** | SIMILARITY > 95% | BREAST, COLON, PANCREAS AND HEAD-NECK CANCERS | 3 |
| | SIMILARITY < 5% | | 509 |
| **OCURRENCCE 10%** | SIMILARITY > 95% | BREAST, COLON, PANCREAS AND HEAD-NECK CANCERS | 3 |
| | SIMILARITY < 5% | | 86 |

**Fig 13. The number of unique triplets identified in each case was analyzed.** These triplets represent the binding of a new protein to a potential drug, characterized by sharing at least one pattern with the original protein.

Fig 12 shows the statistical results of the different similarity metrics. The box plot compares 'Treatment_lung' with the other datasets, with lines ending in a circle. In contrast, the lines ending in squares represent the comparison between 'Full_protein_lung' and the other cancer types.

### 3.4. Drug repurposing

This study has explored innovative computational strategies to identify potential cases for drug repurposing, leading to the creation of several lists that highlight promising repurposing candidates for each disease dataset analyzed (breast, colon, pancreas, head and neck cancers). Fig 13 provides a detailed overview of these results, showcasing each case based on occurrence value. This indicated where patterns were detected in the proteins that form the new triplet, along with the similarity values shared between proteins.

All potential cases identified for each disease will be made available to the scientific community on GitLab https://medal.ctb.upm.es/internal/gitlab/b.otero/lung_cancer_finding_patterns. Upon closer examination of cases with similarity values greater than 95%, it was observed that most had been previously studied, with scientific literature supporting our new associations. Conversely, cases with lower similarity values revealed potential novel repurposing opportunities. Fig 14 illustrates a case identified by sharing five patterns, with a similarity value of less than 1%. This example emphasizes the importance of searching for new repurposing cases based on common patterns, regardless of similarity. The case has been validated in scientific literature as a previous drug repurposing instance.

### 4. Discussion

The relationship between protein structure and function is crucial for understanding disease mechanisms and developing targeted therapies [51]. Advances in computational biology now enable detailed protein sequence analysis, facilitating the identification of functional patterns linked to disease, particularly relevant for complex cancer pathways in oncology and drug discovery [52,53]. This study focuses on developing computational methods to identify critical patterns in protein sequences, especially for non-small cell lung cancer drug targets.

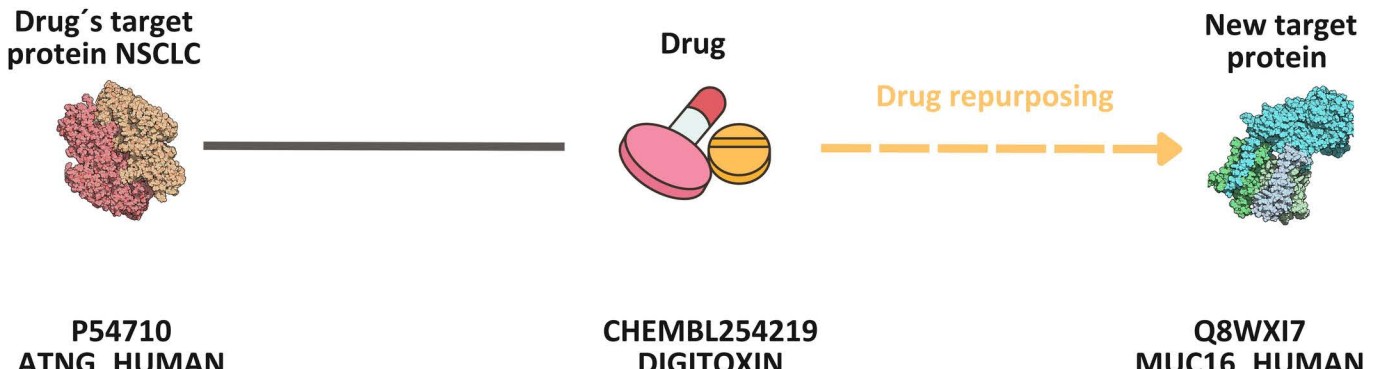

**Fig 14. The method developed in this research identified a potentially successful case of drug repurposing, revealing that these two proteins share at least one common pattern despite exhibiting low similarity.**

Our methodology includes key components, such as a similarity detection method that uses alignment to analyze protein sequence behavior when weights vary. This adjustment enhances data extraction by reducing alignment specificity. Tools like BLAST [54] offer rapid sequence alignment by sacrificing some accuracy for efficiency, while algorithms like Needleman-Wunsch and Smith-Waterman [55] differ in their global versus local alignment approaches. Needleman-Wunsch was selected for this study due to its global orientation, enhancing relationship quality in protein patterns.

We developed algorithms to identify amino acid patterns significant to NSCLC drug-related proteins, leveraging sequence-based relationships without data transformation. This approach offers an alternative to alignment for discovering lexicographic relationships. By integrating alignment and symbolic analysis, it introduces harmonic patterns within sequences [56]. Pattern analysis, typically used for lexicographic issues, is applied here to evaluate symbol distributions rather than the biological complexity. This reduces alignment noise by filtering out non-essential information, although it may omit some interaction details. The combined similarity and pattern-based method reduces collision probability from alphabet repetition alone [57].

Our approach extends beyond non-small cell lung cancer (NSCLC) to explore shared patterns among drug target proteins across various cancer types. By identifying these common patterns. We aim to uncover new uses for lung cancer drugs in treating other cancers. To validate our computational insights, we review scientific literature to find biological support for the identified relationships. This comprehensive strategy enhances our understanding of NSCLC at the molecular level and provides a promising framework for drug repurposing in oncology.

In this study, we introduce a new method for detecting patterns such as recurring amino acid sequences across proteins, using amino acid similarity and occurrence across different proteins. We found that NSCLC drug target proteins share a higher degree of similarity compared to other datasets, and statistically significant differences emerge when comparing NSCLC proteins with their drug targets versus proteins from other cancers. This indicates that similarity is key to identifying functional or family relationships between proteins. The extracted patterns allow us to refine sequence alignment by enabling multi-protein comparisons. By focusing on the longest patterns, we minimize redundancy and noise, as shorter patterns often lack functional relevance due to their increased occurrence among unrelated proteins.

Less frequent patterns tend to be more specialized, making them potentially valuable for targeted drug repurposing. This approach enhances similarity calculations, verifying that these patterns are at least as informative as traditional sequence alignment. By involving more proteins, we obtain broader insights into sequence relationships across all proteins analyzed. Supporting this, our findings show that 3D structures of proteins sharing these patterns align with observed results, as elaborated below.

Lung cancer remains a leading cause of death, with approximately 2.2 million new cases and 1.8 million deaths annually [2]. Although multiple factors contribute to its incidence, smoking control is vital for prevention. Despite extensive molecular research, further investigation into protein-level mechanisms is needed, particularly in examining the 3D structures of involved proteins to find new therapeutic targets. Advances in protein folding prediction have allowed for more detailed structural analyses [42,43]. The identified patterns were further examined by comparing the 3D structures of proteins containing these patterns across different cancers.

The pattern 'CEGCKGFF' appeared in 10% of lung cancer cases, specifically in the proteins RARB and VDR, and was also found in NR1I2 and PPARG within the treatment dataset (Fig 10). Analysis extended to four other cancer types, with the pattern detected at the same 10% frequency (Fig 11a). Nuclear receptors, such as PPARs and TRs, regulate gene expression in response to specific hormones and influence biological functions through complex interactions. Recent research highlights three main mechanisms by which PPARs and TRs interact to regulate gene expression: competition for hormone response element (HRE) binding, context-sensitive DNA binding, and reciprocal regulation. Though these interactions can be cooperative or antagonistic, antagonism is more common. Indirect pathways, such as enzyme gene modulation in T4 metabolism, also play a role in this crosstalk, which is crucial for understanding lipid metabolism and carcinogenesis. Other cellular processes affected by PPAR-TR interaction require further study [58]. Recent studies on PPARG signaling examine connections between obesity, cancer, and the vitamin D/VDR system, introducing new research hypotheses in this area [59]. Additional research investigates vitamin D's role in cancer prevention, specifically analyzing VDR, CYP27B1, CYP24A1, and ROR expression in the human uveal tract and melanoma. This research found an inverse correlation between melanin levels in uveal melanoma and VDR expression, suggesting vitamin D metabolism's role in melanoma and offering potential diagnostic and therapeutic insights for uveal tract disorders. Larger cohort studies are recommended to validate these findings [60]. Nuclear receptors (NRs) are critical in lung cancer development, with studies highlighting the therapeutic potential of small molecules targeting these receptors. As transcription factors, NRs regulate various biological processes, and their dysregulation caused by mutations, epigenetic changes, and altered signaling pathways have been linked to cancer. Research on small molecules targeting specific NRs shows promise in inhibiting tumor growth and inducing apoptosis, underscoring the potential of RN-targeted small molecules as lung cancer therapies [61]. In breast cancer, elevated Retinoid X Receptor Gamma (RXRG) levels in tumor cells correlate with better outcomes, especially in estrogen receptor (ER)-positive cases, suggesting RXRG as a prognostic marker for ER-positive breast cancer patients [62].

The 'QKCL' pattern was observed with a 10% occurrence rate (Fig 11 b, c). Bexarotene, branded as Targretin, is an FDA-approved antineoplastic agent for Cutaneous T cell lymphoma (CTCL), working through a dual mechanism: it inhibits T cell proliferation and induces apoptosis via the p53/p73-dependent pathway [63]. Used off-label for lung and breast cancer and Kaposi's sarcoma, bexarotene binds to RXRs (RXRα, RXRβ, and RXRγ), activating these ligand-activated transcription factors to regulate gene expression. Though its precise mechanism in CTCL remains unclear, bexarotene shows efficacy across all CTCL stages by modulating cell growth, apoptosis, and differentiation. Troglitazone, an insulin-sensitizing agent, was initially used to treat Type II diabetes by activating PPAR-γ receptors, which regulate genes involved in glucose and fatty acid metabolism. It was withdrawn from the market in 2000 due to hepatotoxicity concerns and was replaced by pioglitazone and rosiglitazone. Currently, PPAR-γ is seen as a potential target for chemopreventive therapy to inhibit breast cancer growth [64]. Vismodegib, a hedgehog (Hh) pathway inhibitor, is used to treat locally advanced or metastatic basal cell carcinoma (BCC) by blocking the activity of smoothened homolog (SMO), a key protein in the Hh pathway [65]. Unregulated Hh pathway activation has been linked to cancers including BCC, medulloblastoma, pancreatic cancer, breast cancer, and small cell lung carcinoma, making it a prominent focus in cancer treatment. Research efforts continue to develop precise inhibitors targeting components of the Hh pathway [66]. The NR4A3 nuclear receptor subfamily (NR4A1, NR4A2, and NR4A3) has recently gained recognition as master regulators of cellular processes across multiple organs and diseases. Structurally, these receptors feature an N-terminal AF-1 transactivation

domain, a conserved DNA-binding domain (DBD), and a C-terminal ligand-binding domain (LBD) [67]. NR4A receptors respond to various stressors and play essential roles in maintaining cellular homeostasis and in disease progression. Reduced NR4A expressions in hematological malignancies, such as leukemia and lymphoma, can lead to leukemia onset, whereas increased NR4A expression in solid tumors supports tumor proliferation. Compounds that elevate NR4A can induce apoptosis in hematological cancers, while agents that trigger NR4A1 export from the nucleus can induce apoptosis in solid tumors. Synthetic agents targeting NR4A receptors offer potential as inducers for hematological cancers and antagonists for solid tumors, representing a promising direction for cancer therapeutics [68]. An analysis of nuclear receptors (NRs) in cancer using data from The Cancer Genome Atlas (TCGA) and RNA sequencing on 8,526 samples from 33 cancer types, revealed that while NRs are often under expressed in cancer, some cancers, particularly gynecologic, urologic, and gastrointestinal, showed moderate to high NR expression. Positive correlations in Class IV NR expression were consistent across several cancers. The expression of NRs notably influenced survival rates, with at least five NRs found to have a prognostic value per cancer type. These findings highlight the intricate NR transcriptional networks in cancer and their potential as cancer-specific therapeutic targets [69]. The orphan nuclear receptor NR4A is especially significant in cancer biology due to its absence of known natural ligands, acting as a key regulator connecting inflammation and cancer by modulating gene activity. NR4A functions as a molecular switch influenced by multiple signaling pathways. Impacting both nuclear and mitochondrial function and affecting chemotherapy response in various cancers. Emerging drugs targeting NR4A offer promising new approaches for cancer therapy [70]. Nuclear receptors, fundamental in gene regulation, contain multiple domains that interact with DNA. For instance, peroxisome proliferator-activated receptors (PPARs) form heterodimers with retinoid X receptors (RXRs). PPAR-γ, important for insulin sensitivity, shows therapeutic potential. Recent studies provide insight into the entire structure of PPAR-γ and RXR-α heterodimers bound to DNA, ligands, and coactivator peptides, which form an asymmetric complex allowing the PPAR-γ ligand-binding domain (LBD) to interface with other domains I both proteins. This LBD works with DNA-binding domains (DBDs) to strengthen response element binding. Notably, the A/B segments remain dynamic and unstructured, contributing to gene activation despite their lack of defined structure [71].

The identification of the 'AALET' pattern with a 5% occurrence rate points to important implications for structural analysis (Fig 11d). The human aldehyde dehydrogenase (ALDH) family consists of 19 enzymes that are vital for detoxifying and metabolizing aldehyde substrates. Mutations in ALDH genes can lead to toxic aldehyde buildup, causing metabolic issues and disease. Ethnicity-specific screening for common ALDH alleles can help identify potential risk factors, including cancer susceptibility. Ongoing research into ALDH variants aims to clarify their functional domains, which could deepen our understanding of ALDH enzymes and their roles in human health across different species [72]. Aldehyde dehydrogenase 3 (ALDH1A3), mitochondrial aldehyde dehydrogenase (ALDH2), and mTOR are essential for several biological processes. ALDH1A3 participates in retinoid metabolism, ALDH2 in alcohol metabolism, and mTOR in cell growth and metabolic regulation. Together, they support complex pathways crucial for cellular homeostasis. Notably, ALDH1A3 has emerged as a key metabolic target in cancer diagnosis and therapy, as metabolic reprogramming is central to cancer initiation, metastasis, and recurrence. ALDH1A3 specifically converts all-trans-retinal to retinoic acid, impacting physiological processes and cancer stem characteristics, suggesting it as a potential marker for these cells. Targeting ALDH1A3 thus holds promise for advancing cancer diagnostics and treatment strategies [73]. Disulfiram, an FDA-approved treatment for chronic alcoholism, inhibits ALDH2, disrupting ethanol metabolism and causing acetaldehyde buildup, which triggers adverse symptoms. Disulfiram also inhibits ALDH1A1, an enzyme involved in converting retinal to retinoic acid, which plays significant roles in conditions like cancer and obesity. Derivative 2b was synthesized to act similarly on ALDH1A1 without inhibiting ALDH2, suggesting selective inhibition potential [74]. Disulfiram also shows promise as an anti-cancer agent by inducing cellular stress, overcoming drug resistance, and reducing angiogenesis, particularly when combined with copper [75]. Everolimus, an mTOR inhibitor, has demonstrated antiproliferative and antiangiogenic effects in solid tumours and immunosuppressive properties, providing affective in T-cell lymphoma (TCL) with a 44% response rate in relapsed patients [76].

Novel computational methodologies were applied in this study to identify potential drug repurposing opportunities. These approaches led to the generation of curated lists of promising repurposing candidates for each of the disease-specific datasets analyzed, including those related to breast, colon, pancreatic, and head and neck cancers.

Upon closer examination of cases with similarity values greater than 95%, it was observed that most had been previously studied, with scientific literature supporting our new associations. A notable example is the drug vinblastine, which is used as a therapeutic agent for non-small cell lung cancer (NSCLC) and is also employed in breast cancer treatment. Our methodology identified a significant pattern and a similarity of over 95%, establishing a correlation between this drug and both diseases [77,78]. This finding validates the utility of our novel approach to identify repurposing cases. Conversely, cases with lower similarity values revealed potential novel repurposing opportunities. Fig 14 illustrates a case identified by sharing five patterns, with a similarity value of less than 1%. This example emphasizes the importance of searching for new repurposing cases based on common patterns, regardless of similarity. The case has been validated in scientific literature as a previous instance of drug repurposing. Originally developed to treat congestive cardiac insufficiency, arrhythmias, and heart failure, its potential as an anticancer treatment is evident, supporting its association with lung cancer as a repurposing case. The relationship identified through pattern searching supports the use of this drug with the respective target protein across various cancer types. Numerous studies corroborate its application in lung cancer, serving as our starting point, along with its broader anticancer function. Additionally, there is substantial evidence for its use in breast cancer [79] and colorectal cancer [80], among others.

Utilizing significant patterns identified in the target proteins of lung cancer drugs as a foundation for exploring new drug repurposing opportunities appears to be a promising strategy with substantial potential. By applying this methodology to various datasets, new triplets have been discovered that connect the original target protein associated with non-small cell lung cancer (NSCLC) to other cancer types via the drugs used in NSCLC. The lists provided may be invaluable for initiating new projects to uncover the extent of these new relationships that could lead to successful outcomes. The computationally identified cases can be validated through scientific literature, as has been done with several instances, or they can progress to clinical trials in a wet lab setting for in vitro or in vivo testing.

By detecting common patterns between proteins, we can identify signatures that help detect proteins with shared features, enhancing our understanding of disease mechanisms. The potential impact of this work extends beyond lung cancer, offering a new paradigm for drug discovery that could significantly reduce the time and cost associated with making new cancer treatments available to patients. As we continue to refine and expand our methodologies, we anticipate that this approach will contribute to the development of more personalized and effective cancer therapies.

## 5. Conclusions

In summary, our research presents a robust computational methodology for identifying significant protein patterns in disease-related sequences, revealing potential therapeutic targets for conditions such as cancer and autoimmune disorders. This approach also enables the exploration of complex protein relationships across diseases and introduces an innovative strategy for drug repurposing, accelerating treatment discovery.

The detection of patterns in amino acid sequences to identify similar proteins and predict their 3D structures has applications in various fields, including target identification, antibody design, enzyme development, and protein engineering. It plays a crucial role in drug discovery, enabling drug repurposing and therapeutic design, as well as in biotechnology and synthetic biology, where protein functionality can be modified or enhanced.

Future work will focus on expanding the methodology to additional diseases, assessing the biological relevance of identified patterns, and visualizing 3D structures to better understand their role in drug interactions. These findings may also inform new clinical trials, further advancing therapeutic development.

While there are inherent limitations in any in silico approach, such as the absence of drug interaction data, pathway alterations, and expression information, this study establishes a solid foundation for understanding protein relationships and guiding further in this field.

## Supporting Information

**S1 Figure. Histograms show the distribution of the four protein similarity metrics across the different datasets.**
(DOCX)

**S1 File. Methods.** Likelihood of subsequence occurrence and threshold determination.
(DOCX)

**S1 Table. Shows these drugs together with their target protein, the derived gene, and the effect of the drug on the disease.**
(DOCX)

## Author contributions

**Conceptualization:** Belén Otero-Carrasco, Aurora Peréz Peréz.

**Data curation:** Belén Otero-Carrasco, Gema Díaz Ferreiro.

**Formal analysis:** Belén Otero-Carrasco, Gema Díaz Ferreiro.

**Funding acquisition:** Alejandro Rodríguez-González.

**Investigation:** Belén Otero-Carrasco, Paloma Tejera Nevado.

**Methodology:** Belén Otero-Carrasco, Paloma Tejera Nevado.

**Software:** Rafael Artiñano Muñoz, Gema Díaz Ferreiro.

**Supervision:** Aurora Peréz Peréz, Juan Pedro Caraça-Valente Hernández, Alejandro Rodríguez-González.

**Validation:** Belén Otero-Carrasco, Paloma Tejera Nevado, Rafael Artiñano Muñoz.

**Visualization:** Belén Otero-Carrasco.

**Writing – original draft:** Belén Otero-Carrasco.

**Writing – review & editing:** Belén Otero-Carrasco, Paloma Tejera Nevado, Rafael Artiñano Muñoz, Aurora Peréz Peréz, Juan Pedro Caraça-Valente Hernández, Alejandro Rodríguez-González.

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
