## [Decision Letter · Decision Letter 0]

31 Jan 2025

PONE-D-24-57595Finding patterns in lung cancer protein sequences for drug repurposingPLOS ONE

Dear Dr. Otero Carrasco,

Thank you for submitting your manuscript to PLOS ONE. After careful consideration, we feel that it has merit but does not fully meet PLOS ONE’s publication criteria as it currently stands. Therefore, we invite you to submit a revised version of the manuscript that addresses the points raised during the review process.

**ACADEMIC EDITOR: **

After carefully considering the reviews and assessing your manuscript, I would like to invite you to revise and resubmit your manuscript for further consideration. The reviewers have provided constructive comments that will help strengthen your work. Please address each of these points thoroughly in your revised manuscript. Additionally, ensure that you provide a detailed response letter outlining how you have addressed each comment raised by the reviewers. This will help the reviewers and myself to evaluate the changes made to the manuscript.  Any additional references suggested during the peer-review process should only be included if the authors agree that they are relevant and useful.

We look forward to receiving your revised manuscript.

Kind regards,

Khalid Raza, PhD (Computational Biology)

Academic Editor

PLOS ONE

Journal requirements: When submitting your revision, we need you to address these additional requirements. 1. Please ensure that your manuscript meets PLOS ONE's style requirements, including those for file naming. The PLOS ONE style templates can be found at https://journals.plos.org/plosone/s/file?id=wjVg/PLOSOne_formatting_sample_main_body.pdf and https://journals.plos.org/plosone/s/file?id=ba62/PLOSOne_formatting_sample_title_authors_affiliations.pdf. 2. Please note that PLOS ONE has specific guidelines on code sharing for submissions in which author-generated code underpins the findings in the manuscript. In these cases, we expect all author-generated code to be made available without restrictions upon publication of the work. Please review our guidelines at https://journals.plos.org/plosone/s/materials-and-software-sharing#loc-sharing-code and ensure that your code is shared in a way that follows best practice and facilitates reproducibility and reuse. 3. We note that the grant information you provided in the ‘Funding Information’ and ‘Financial Disclosure’ sections do not match.  When you resubmit, please ensure that you provide the correct grant numbers for the awards you received for your study in the ‘Funding Information’ section. 4. Thank you for stating the following financial disclosure:  [This research was funded by the project “Data-driven drug repositioning applying graph neural networks (3DR-GNN)” (PID2021-122659OB-I00) from the Spanish Ministerio de Ciencia e Innovación, “Drug repurposing hypotheses through a data-driven approach (GRENADA)” (PDC2022-133173-I00) from the Spanish Ministerio de Ciencia e Innovación and MadridDataSpace4Pandemics, funded by Comunidad de Madrid (Consejería de Educación, Universidades, Ciencia y Portavocía) with FEDER funds as part of the response from the  European Union to COVID-19 pandemia. Belen Otero Carrasco’s work is supported by “Formación de Personal Investigador” grant (FPI PRE2019-090912) as part of the project “DISNET (Creation and analysis of disease networks for drug repurposing from heterogeneous data sources applied to rare diseases)” (RTI2018-094576-A-I00) from the Spanish Ministerio de Ciencia, Innovación y Universidades.].  Please state what role the funders took in the study.  If the funders had no role, please state: ""The funders had no role in study design, data collection and analysis, decision to publish, or preparation of the manuscript."" If this statement is not correct you must amend it as needed. Please include this amended Role of Funder statement in your cover letter; we will change the online submission form on your behalf. 5. We note that your Data Availability Statement is currently as follows: [All relevant data are within the manuscript and its Supporting Information files.] Please confirm at this time whether or not your submission contains all raw data required to replicate the results of your study. Authors must share the “minimal data set” for their submission. PLOS defines the minimal data set to consist of the data required to replicate all study findings reported in the article, as well as related metadata and methods (https://journals.plos.org/plosone/s/data-availability#loc-minimal-data-set-definition). For example, authors should submit the following data: - The values behind the means, standard deviations and other measures reported;- The values used to build graphs;- The points extracted from images for analysis. Authors do not need to submit their entire data set if only a portion of the data was used in the reported study. If your submission does not contain these data, please either upload them as Supporting Information files or deposit them to a stable, public repository and provide us with the relevant URLs, DOIs, or accession numbers. For a list of recommended repositories, please see https://journals.plos.org/plosone/s/recommended-repositories. If there are ethical or legal restrictions on sharing a de-identified data set, please explain them in detail (e.g., data contain potentially sensitive information, data are owned by a third-party organization, etc.) and who has imposed them (e.g., an ethics committee). Please also provide contact information for a data access committee, ethics committee, or other institutional body to which data requests may be sent. If data are owned by a third party, please indicate how others may request data access. 6. Please upload a copy of Figure S1 to which you refer in your text on page 34. Please amend the file type to 'Supporting Information'. If the Supplementary file is no longer to be included as part of the submission please remove all reference to it within the text.

Reviewers' comments:

Reviewer's Responses to Questions

**Comments to the Author**

1. Is the manuscript technically sound, and do the data support the conclusions?

Reviewer #1: Yes

Reviewer #2: Yes

Reviewer #3: Yes

2. Has the statistical analysis been performed appropriately and rigorously? 

Reviewer #1: Yes

Reviewer #2: Yes

Reviewer #3: Yes

3. Have the authors made all data underlying the findings in their manuscript fully available?

Reviewer #1: Yes

Reviewer #2: Yes

Reviewer #3: No

4. Is the manuscript presented in an intelligible fashion and written in standard English?

Reviewer #1: Yes

Reviewer #2: Yes

Reviewer #3: Yes

5. Review Comments to the Author

Reviewer #1: The study is a novel approach of computational methods to identify protein sequence patterns for drug repurposing, specially for NSCLC treatment. The work integrated data from sources, such as UniProt and DISNET, to compile and for analysis and reproducibility. This the study provides new idea for drug repurposing and to understand protein functionality. Further, validation through 3D structural analysis using AlphaFold and ChimeraX adds robustness to the findings, to bridge the computational and structural biology. The step-by-step description of pattern discovery, statistical analysis, and similarity metrics shows the clarity.

However the limitations include: 1) No laboratory experiments to validate. 2) Sensitivity analysis for varying thresholds is not discussed 3) Missing statistical justification for pattern selection, especially the minimum length of four amino acids and occurrence rates 4) The RMSD there is no detailed explanation of how deviations impact biological functionality.

Reviewer #2: 1. How did the amino acid sequences for each lung cancer drug target protein get

gathered in order to find important trends in the protein sequences of treatments?

2. Why was the DISNET database picked to store the amino acid sequences of proteins

that are used to treat lung cancer? What does DISNET do better than other databases?

3. Refine the loops of the proteins shown in Figures 10 and 11.

4. The RMSD values are usually between 0–2Å, but Table 3 shows unusually high

RMSD values for the target structures compared to the reference.

5. Only one amino acid pattern, AALET, has an RMSD score of 4.287.

6. Increase the resolution of Figure 12 to 300 dpi.

7. Update the graphical abstract to better reflect the key points of the manuscript.

8. Add a clear and concise takeaway message in the conclusion section.

9. Use a white background for the visuals.

10. After applying the 10% occurrence level, there are only 2,034 unique patterns, which

is a lot less than the previous case, even though there are 14,330 pattern-protein

combinations.

11. How were the patterns that showed up in proteins linked to NSCLC that were 5% and

10% of the time in lung cancer medicines looked at to find out where they were

found? How was this analysis done? What factors or methods were used?

12. How were the 2,200 patterns found in proteins of different types of cancer, out of the

2,368 patterns found with a 5% chance of occurring in lung cancer treatments? What

method or approach was used to find out if they were there?

Reviewer #3: General comment

The authors propose a computational analysis for identifying potentially drugs candidate for non-small cells lung cancer based on the identification of characteristic patterns that link the drug’s target proteins to the proteins associated to lung cancer. These distinctive patterns of lung cancer drug target proteins were then searched for in other diseases-associated proteins. The manuscript is well-written, scientifically sound, and addresses a relevant topic to the scientific community, but some improvements and clarifications need to be performed to the presentation of the study, especially advances with respect to other DR approaches.

Major Comments

1. The study described in the manuscript is really interesting, but the novelty of the method should be better highlighted. The authors should better emphasize the significant advance of this study within the field of network-based strategies for drug repurposing opportunities with respect to other previous published ones about the same topic. In particular, the authors should deeply discuss the: (1) the rationale behind the idea of using disease-related protein sequences and of this approach (2) the advances of their approach respect to the framework of well-established Network Medicine to find repurposable drugs applied for study for example COVID-19 here https://www.pnas.org/doi/10.1073/pnas.2025581118 and here https://doi.org/10.1371/journal.pcbi.1008686, and in human cancer here: https://www.sciencedirect.com/science/article/pii/S2001037022005311

Moreover, these recent studies about successful network-based approaches for drug repurposing applied to study complex diseases, should be cited (for example at line 77, when talking about computational approaches for DR):

o Network medicine framework for identifying drug-repurposing opportunities for COVID-19 PNAS May 11, 2021 118 (19) e2025581118; https://doi.org/10.1073/pnas.2025581118

o Comprehensive network medicine-based drug repositioning via integration of therapeutic efficacy and side effects. npj Syst Biol Appl 8, 12 (2022). https://doi.org/10.1038/s41540-022-00221-0

o SAveRUNNER: an R-based tool for drug repurposing. BMC Bioinformatics 22, 150 (2021). https://doi.org/10.1186/s12859-021-04076-w

o Drug Repurposing: A Network-based Approach to Amyotrophic Lateral Sclerosis. Neurotherapeutics. 2021 Jul;18(3):1678-1691. doi: 10.1007/s13311-021-01064-z. Epub 2021 May 13. PMID: 33987813; PMCID: PMC8609089.

2. The authors could exploit the Connectivity Map (CMAP) database (https://clue.io/) to query for a core set of 9 different cell lines (at https://clue.io/query) in which the compounds and genetic perturbagens have been profiled and to see if the identified drug could conteract the action of the disease signature.

3. All figures have to be improved. Just few examples:

- Fonts are too small and barely readable (e.g. Figure 6, Figure 9) or label appear stretched (Figure 13)

- The number of Figure is too high, I suggest to move som figure as supplementary figures

4. The authors need to include a final conclusion at the end of the manuscript and also discuss about the limitations of their approach.

Minor

1. In general, the authors should check the form of the entire paper avoiding careless errors, missing commas, missing space between words.

2. Abstract is too technical, please avoid acronyms and details that are not useful for fluent and clear reading. Moreover, first sentence is too long, please break it into two sentences or re-formulate it.

3. Check all the commas. Add a comma before “and” whenever more than two elements have been listed.

4. Figure 3’s legend is too long

6. PLOS authors have the option to publish the peer review history of their article (what does this mean? ). If published, this will include your full peer review and any attached files.

**Do you want your identity to be public for this peer review?** For information about this choice, including consent withdrawal, please see our Privacy Policy .

Reviewer #1: No

Reviewer #2: No

Reviewer #3: No

---

## [Author Response · Author response to Decision Letter 1]

10 Mar 2025

Subject: Response to Reviewers’ Comments on Manuscript [PONE-D-24-57595]

Dear Dr. Khalid Raza,

We appreciate the time and effort that you and the reviewers have taken to evaluate our manuscript titled “Finding patterns in lung cancer protein sequences for drug repurposing”. We are grateful for the constructive feedback which has helped us improve the quality of our work.

We have carefully revised the entire manuscript to address the reviewer’s comments regarding punctuation, spacing, and overall clarity. During this process, we identified some sentences that could be unclear and have improved their readability accordingly. These modifications have been highlighted in the revised manuscript.

Additionally, given that we received questions regarding the selection of the pattern length in the analysis and the occurrence values, we have now included a detailed explanation and calculations in the Supporting Information. We believe this addition will be valuable for readers who may have similar questions. Below, we provide our responses to the editor and reviewer’s comments. All changes in the revised manuscript have been highlighted for easy reference.

Thank you for your comments. We have carefully reviewed the points raised. Please let us know if any additional information or modifications are needed.

Regarding point one, we have carefully reviewed the manuscript and have ensured full compliance with all of PLOS ONE's style guidelines, including the correct application of file naming conventions. Additionally, we have taken the necessary steps to remove any zip codes as per the journal's requirements. Furthermore, we have updated the manuscript to include all requested information for the corresponding author, ensuring that every detail aligns with the submission guidelines.

2. Please note that PLOS ONE has specific guidelines on code sharing for submissions in which author-generated code underpins the findings in the manuscript.

We would like to confirm that our code and related files have been made publicly available in a repository, ensuring unrestricted access for the scientific community. This ensures full reproducibility and facilitates reuse, fully in accordance with PLOS ONE’s code-sharing guidelines.

We have reviewed and corrected the grant information to ensure consistency between the ‘Funding Information’ and ‘Financial Disclosure’ sections. The correct grant number have now been provided.

[This research was funded by the project “Data-driven drug repositioning applying graph neural networks (3DR-GNN)” (PID2021-122659OB-I00) from the Spanish Ministerio de Ciencia e Innovación, “Drug repurposing hypotheses through a data-driven approach (GRENADA)” (PDC2022-133173-I00) from the Spanish Ministerio de Ciencia e Innovación and MadridDataSpace4Pandemics, funded by Comunidad de Madrid (Consejería de Educación, Universidades, Ciencia y Portavocía) with FEDER funds as part of the response from the European Union to COVID-19 pandemia. Belen Otero Carrasco’s work is supported by “Formación de Personal Investigador” grant (FPI PRE2019-090912) as part of the project “DISNET (Creation and analysis of disease networks for drug repurposing from heterogeneous data sources applied to rare diseases)” (RTI2018-094576-A-I00) from the Spanish Ministerio de Ciencia, Innovación y Universidades.].

Thank you for your comments. We have amended the financial disclosure statement as requested. We have now added the appropriate statement regarding the role of the funders in the study. The revised statement has been included in the cover letter as per your instructions.

5. We note that your Data Availability Statement is currently as follows: [All relevant data are within the manuscript and its Supporting Information files.]

Please confirm at this time whether or not your submission contains all raw data required to replicate the results of your study. Authors must share the "minimal data set" for their submission. PLOS defines the minimal data set to consist of the data required to replicate all study findings reported in the article, as well as related metadata and methods (https://journals.plos.org/plosone/s/data-availability#loc-minimal-data-set-definition).

We have reviewed the Data Availability Statement and have updated it accordingly. The relevant data is now available and has been deposited in a public repository, ensuring that the minimal data set required to replicate the study’s findings is accessible.

6. Please upload a copy of Figure S1 to which you refer in your text on page 34. Please amend the file type to 'Supporting Information'. If the Supplementary file is no longer to be included as part of the submission, please remove all reference to it within the text.

We have uploaded a copy of Figure S1, as referenced in the text on page 34. Additionally, we have also uploaded the Supporting Information Methods S1 and Table S1. Both files have been correctly marked as “Supporting Information”.

Reviewers

Reviewer #1: The study is a novel approach of computational methods to identify protein sequence patterns for drug repurposing, specially for NSCLC treatment. The work integrated data from sources, such as UniProt and DISNET, to compile and for analysis and reproducibility. This the study provides new idea for drug repurposing and to understand protein functionality. Further, validation through 3D structural analysis using AlphaFold and ChimeraX adds robustness to the findings, to bridge the computational and structural biology. The step-by-step description of pattern discovery, statistical analysis, and similarity metrics shows the clarity. However the limitations include:

R1 – C1: No laboratory experiments to validate.

Thank you for this important point. This study is based on a fully computational approach, which serves as a crucial preliminary step before experimental validation. By refining our methodology in this phase, we aim to generate valuable insights that can guide future laboratory experiments. While direct experimental validation is not included in the present work, we recognize its importance and are actively initiating collaborations with researchers who have conducted relevant laboratory experiments in this area. These collaborations will enable us to explore whether our hypotheses align with experimental findings, thereby enhancing the robustness and applicability of our results. Additionally, the data and results presented in this work can serve as a valuable resource for other researchers conducting similar analyses, facilitating knowledge exchange and future experimental studies.

R1 – C2: Sensitivity analysis for varying thresholds is not discussed

Thank you for your valuable comment. We understand sensitivity analysis as the process of testing how changes in key parameter (in this case, thresholds), impact the outcome. We address this explanation of the thresholds used, 5% and 10% in comment R1-C3. If the comment refers to the similarity calculations, we would like to clarify that the four metrics were computed, and the statistical analysis was conducted following the establish methodology in this field.

R1 – C3: Missing statistical justification for pattern selection, especially the minimum length of four amino acids and occurrence rates.

In the context of our research, a pattern is a common subsequence that appears in a significant number of sequences within a group. In this study, we propose identifying patterns within a group of NSCLC treatment protein sequences and subsequently checking whether any of these patterns appear in protein sequences associated with other diseases. Due to natural variations in proteins, patterns can arise by chance. However, this randomness diminishes when considering related proteins that share common structural and functional traits [Ref 1].

The total number of possible subsequences is proportional to the subsequence size (length) and the number of available elements for combination. In protein sequences, these elements correspond to the 20 basic amino acids, where each amino acid can occupy any position within the sequence. The number of possible distinct subsequences of these elements is computed as a variation with repetition and is given by Equation 1, where x denotes the length of the subsequence:

VR_(20,x)=20^x (1)

As x increases, the likelihood of identifying a recurring pattern decreases, meaning that longer subsequences are less likely to appear in a sufficiently high number of proteins. Given that amino acid combinations are assumed to be equally likely, the likelihood of a given subsequence is 1/20^x . Additionally, these subsequences can emerge at any position within a protein sequence. Accordingly, to test whether a given subsequence of length x appears in a protein of length protein_length, considering that any position up to the last (x-1) is a potential place for it to be located, the number of chances for that subsequence to occur is therefore protein_length - (x - 1).

The choice of a minimum pattern length of four amino acids is based on both probabilistic and biological reasoning. Since protein sequences are composed of 20 amino acids, the number of possible distinct subsequences increases exponentially with length. For example, there are 160,000 possible combinations for sequences of length 4, while for sequences of length 6, this number grows to 6,400,000. Given that amino acid combinations are assumed to be equally likely, shorter patterns have a much higher likelihood of appearing randomly within a dataset.

This effect can be quantified using probability calculations. When analyzing protein sequences, a pattern of length 4 has a significantly higher chance of emerging by chance than longer patterns. In contrast, as the length of the pattern increases, the probability of random occurrence decreases exponentially, making these longer patterns much less likely to appear unless they have functional or structural significance. This is particularly relevant in our dataset, where the treatment set consists of only 52 proteins. In such a small dataset, even detecting patterns of length 4 is already challenging unless these patterns are genuinely shared among proteins rather than appearing due to random chance.

From a biological perspective, shorter patterns tend to carry less specific information about structural or functional properties of proteins. Since amino acids interact in groups to define motifs, domains, and binding sites, patterns of only two or three amino acids may not be sufficient to capture relevant biological features, as they can be part of many unrelated sequences. As pattern length increases, there is a greater likelihood that these sequences correspond to conserved or functionally relevant regions, making them more informative for understanding protein interactions and mechanisms.

For this reason, setting a minimum threshold of four amino acids ensures that the identified patterns are less likely to be artifacts of random sequence variability and increases the probability of capturing biologically relevant motifs. Additionally, previous studies have established that biologically relevant sequence similarities are typically observed for subsequences of at least four amino acids [Ref 2]. This threshold is thus a compromise between statistical robustness and biological interpretability, allowing us to identify meaningful patterns while minimizing the likelihood of coincidental similarities.

The occurrence rates represent the number of proteins that must share a pattern for it to be considered relevant. In this study, we employed 5% and 10% occurrence thresholds, which act as hyperparameters that regulate the balance between specificity and generality in pattern detection. A higher threshold (10%) is more restrictive, leading to the identification of shorter, more conserved patterns, as it requires them to be present in a greater number of proteins. Conversely, a lower threshold (5%) is more permissive, allowing for the detection of longer patterns that appear in fewer proteins, which suggests they may be more selective or associated with specific subgroups within the dataset.

The selection of these thresholds was driven by empirical evaluation, ensuring that the identified patterns were sufficiently long (at least four amino acids) while still being general enough to capture relevant sequence information. Through iterative testing, we observed that 5% and 10% provided an optimal balance between detecting meaningful patterns and avoiding excessive variability. Values above 10% significantly reduced the number of detected patterns, potentially discarding relevant information, while values below 5% introduced too much variability, leading to patterns that were less reproducible or less biologically meaningful. Additionally, three-dimensional structural analysis supported the biological relevance of the identified patterns, particularly for proteins within the same family or functionally related groups.

It is important to emphasize that these thresholds are dataset-dependent and should not be considered fixed for all studies. The optimal occurrence rate must be adjusted based on the nature and size of the dataset to ensure the best balance between specificity and sensitivity. Future applications of this methodology should consider tuning this hyperparameter according to the characteristics of the protein sequences under investigation, as different datasets may require different thresholds to maximize biological interpretability.

To address these concerns, an annex has been created and will be included in the supporting information (Methods S1). The formulas applied to our dataset yielded the results that are provided in the supporting information containing the calculations and tables. This has been provided to the reviewer.

[Ref 1] - Z. Zou & J. Zhang. 2019 Amino acid exchangeabilities vary across the three of life, Sci. Adv., vol. 5, no. 12, p. eaax3124.

[Ref 2] - M. A. Roytber. 1992. A search for common patterns in many sequences. CABIOS, vol. 8, no. 1, pages: 57-64.

R1 – C4: The RMSD there is no detailed explanation of how deviations impact biological functionality.

Thank you for highlighting this point. Various methods are regularly used to compare computational models with experimental data in modeling assessments. Among these, RMSD is the most applied quantitative metric for evaluating the similarity between two superimposed atomic coordinate sets, typically expressed in angstroms. Global RMSD, which is based on positional distance, is often used to assess overall protein structure similarity. The RMSD value represents the average deviation between corresponding atoms of two proteins, with lower RMSD indicating greater structural similarity. The primary drawback of RMSD is that it is heavily influenced by the size of errors. Two structures that are otherwise identical, except for the position of a single loop or a flexible terminus, often show a large global backbone RMSD [Ref 1]. It has been shown through analyses evaluating protein models against native structures that RMSD exhibits the greatest variation. RMSD values are significantly influenced by large errors, while other scores tend to focus on more accurate regions or are inherently local. As a result, RMSD is the least reliable in this context, as incomplete models are more likely to receive better scores [Ref 2]. In the present study, RMSD was used as a metric to evaluate the differences between the protein linked to a drug (treatment) and the cand

---

## [Decision Letter · Decision Letter 1]

25 Mar 2025

Finding patterns in lung cancer protein sequences for drug repurposing

PONE-D-24-57595R1

Dear Dr. Otero Carrasco,

We’re pleased to inform you that your manuscript has been judged scientifically suitable for publication and will be formally accepted for publication once it meets all outstanding technical requirements.

Kind regards,

Khalid Raza, PhD (Computational Biology)

Academic Editor

PLOS ONE

Additional Editor Comments (optional):

Reviewers' comments:

Reviewer's Responses to Questions

**Comments to the Author**

1. If the authors have adequately addressed your comments raised in a previous round of review and you feel that this manuscript is now acceptable for publication, you may indicate that here to bypass the “Comments to the Author” section, enter your conflict of interest statement in the “Confidential to Editor” section, and submit your "Accept" recommendation.

Reviewer #1: All comments have been addressed

Reviewer #3: All comments have been addressed

2. Is the manuscript technically sound, and do the data support the conclusions?

Reviewer #1: Yes

Reviewer #3: Yes

3. Has the statistical analysis been performed appropriately and rigorously? 

Reviewer #1: Yes

Reviewer #3: Yes

4. Have the authors made all data underlying the findings in their manuscript fully available?

Reviewer #1: Yes

Reviewer #3: Yes

5. Is the manuscript presented in an intelligible fashion and written in standard English?

Reviewer #1: Yes

Reviewer #3: Yes

6. Review Comments to the Author

Reviewer #1: Authors have addressed all the comments and answered them effectively. Hence the revised manuscript can be accepted for the publication.

Reviewer #3: The authors addressed all my comments and thus I suggest the acceptance of the manuscript that has been improved

7. PLOS authors have the option to publish the peer review history of their article (what does this mean? ). If published, this will include your full peer review and any attached files.

**Do you want your identity to be public for this peer review?** For information about this choice, including consent withdrawal, please see our Privacy Policy .

Reviewer #1: No

Reviewer #3: No

---

## [Editor Report · Acceptance letter]

PONE-D-24-57595R1

PLOS ONE

Dear Dr. Otero-Carrasco,

I'm pleased to inform you that your manuscript has been deemed suitable for publication in PLOS ONE. Congratulations! Your manuscript is now being handed over to our production team.

Kind regards,

on behalf of

Dr. Khalid Raza

Academic Editor

PLOS ONE